# Interrogation of Essentiality in the Reconstructed *Haemophilus influenzae* Metabolic Network Identifies Lipid Metabolism Antimicrobial Targets: Preclinical Evaluation of a FabH *β*-Ketoacyl-ACP Synthase Inhibitor

Nahikari López-López,[a] David San León,[b] Sonia de Castro,[c] Roberto Díez-Martínez,[d] Manuel Iglesias-Bexiga,[e] María José Camarasa,[c] Margarita Menéndez,[e,f] Juan Nogales,[b,g] Junkal Garmendia[a,f]

[a]Instituto de Agrobiotecnología, Consejo Superior de Investigaciones Científicas (IdAB-CSIC)-Gobierno de Navarra, Mutilva, Spain
[b]Centro Nacional de Biotecnología, Consejo Superior de Investigaciones Científicas (CNB-CSIC), Madrid, Spain
[c]Instituto de Química Médica, Consejo Superior de Investigaciones Científicas (IQM-CSIC), Madrid, Spain
[d]Telum Therapeutics, Noain, Spain
[e]Instituto de Física Química Rocasolano, Consejo Superior de Investigaciones Científicas (IQFR-CSIC), Madrid, Spain
[f]Centro de Investigación Biomédica en Red de Enfermedades Respiratorias (CIBERES), Madrid, Spain
[g]Interdisciplinary Platform for Sustainable Plastics towards a Circular Economy-Spanish National Research Council (SusPlast-CSIC), Madrid, Spain

**ABSTRACT** Expediting drug discovery to fight antibacterial resistance requires holistic approaches at system levels. In this study, we focused on the human-adapted pathogen *Haemophilus influenzae*, and by constructing a high-quality genome-scale metabolic model, we rationally identified new metabolic drug targets in this organism. Contextualization of available gene essentiality data within *in silico* predictions identified most genes involved in lipid metabolism as promising targets. We focused on the *β*-ketoacyl-acyl carrier protein synthase III FabH, responsible for catalyzing the first step in the FASII fatty acid synthesis pathway and feedback inhibition. Docking studies provided a plausible three-dimensional model of FabH in complex with the synthetic inhibitor 1-(5-(2-fluoro-5-(hydroxymethyl)phenyl)pyridin-2-yl)piperidine-4-acetic acid (FabHi). Validating our *in silico* predictions, FabHi reduced *H. influenzae* viability in a dose- and strain-dependent manner, and this inhibitory effect was independent of *fabH* gene expression levels. *fabH* allelic variation was observed among *H. influenzae* clinical isolates. Many of these polymorphisms, relevant for stabilization of the dimeric active form of FabH and/or activity, may modulate the inhibitory effect as part of a complex multifactorial process with the overall metabolic context emerging as a key factor tuning FabHi activity. Synergies with antibiotics were not observed and bacteria were not prone to develop resistance. Inhibitor administration during *H. influenzae* infection on a zebrafish septicemia infection model cleared bacteria without signs of host toxicity. Overall, we highlight the potential of *H. influenzae* metabolism as a source of drug targets, metabolic models as target-screening tools, and FASII targeting suitability to counteract this bacterial infection.

**IMPORTANCE** Antimicrobial resistance drives the need of synergistically combined powerful computational tools and experimental work to accelerate target identification and drug development. Here, we present a high-quality metabolic model of *H. influenzae* and show its usefulness both as a computational framework for large experimental data set contextualization and as a tool to discover condition-independent drug targets. We focus on *β*-ketoacyl-acyl carrier protein synthase III FabH chemical inhibition by using a synthetic molecule with good synthetic and antimicrobial profiles that specifically binds to the active site. The mechanistic complexity of FabH inhibition may go beyond allelic variation, and the strain-dependent effect of the inhibitor tested supports the impact of metabolic context as a key factor driving bacterial cell behavior. Therefore, this study highlights the systematic

Address correspondence to Junkal Garmendia, juncal.garmendia@csic.es.

The authors declare no conflict of interest.

10.1128/msystems.01459-21 **1**

metabolic evaluation of individual strains through computational frameworks to identify secondary metabolic hubs modulating drug response, which will facilitate establishing synergistic and/or more precise and robust antibacterial treatments.

**KEYWORDS** *Haemophilus influenzae*, airway infection, genome-scale metabolic model, gene essentiality screening, fatty acid synthesis, FabH inhibition, antimicrobials, preclinical evaluation

*H*aemophilus influenzae is a human-adapted Gram-negative bacterial pathogen. Asymptomatic colonization begins in the upper airways, but it can spread through the respiratory tract and lead to invasive infections. The polysaccharide capsule vaccine has driven the almost complete disappearance of capsulated *H. influenzae* type b (Hib) in countries with established child immunization programs (1, 2). Instead, other *H. influenzae* serotypes and nontypeable strains (NTHi), untargeted by the Hib vaccine, emerge as important causes of infections. NTHi causes otitis media, conjunctivitis, sinusitis, and lower respiratory infections in children; exacerbations of chronic obstructive pulmonary disease (COPD) and cystic fibrosis (CF) in adults; and invasive disease in neonates, immunocompromised adults, and the elderly (2–5). Ampicillin (Amp)-resistant *H. influenzae* is included in the WHO global priority list of bacteria for which new antibacterials are urgently needed (6), fluoroquinolone resistance has increased in recent years, spreading worldwide with a variety in epidemiology, and the use of the macrolides in severe COPD patients may lead to decreased macrolide susceptibility (7–9). The current antibiotic resistance public health crisis prompted us to evaluate novel therapeutic alternatives against *H. influenzae*.

Bacterial metabolism and the effect of antimicrobial drugs are linked at several levels in such a way that a metabolism-based approach may counteract antibiotic tolerance by the use of exogenous metabolites, inducing antibiotic internalization, inhibiting antibiotic detoxification or extrusion, inducing endogenous oxidative stress, or priming to proton motive force (10, 11). Inactivation of central metabolic enzymes may contribute to develop either antibiotic susceptibility or resistance (12–14). One strategy to develop effective antimicrobial treatments is to identify and develop inhibitors of essential biological pathways, and bacterial metabolism is a source of drug targets, as metabolic genes can be unconditionally essential. In this context, genome-scale network reconstructions of bacterial metabolism can serve as computational test-bed platforms to identify metabolic enzymes as potential targets (15–17). Such reconstructions are structured, species-specific knowledge bases that contain detailed information on the target organism, such as the exact reaction stoichiometry and reversibility, relationships between genes, proteins, and reactions, and the biochemical and physiological data available at the time of the reconstruction. These reconstructions can be further converted into genome-scale models (GEMs) that enable computation of the metabolic capabilities and phenotype of a given organism (18).

*H. influenzae* was the first organism for which a GEM was constructed, just shortly after genome sequencing (19). This pioneer GEM laid the foundations of the field and was used to study minimal substrate requirements for the network to allow biomass production (19–22). However, this initial model is far from the increasing metabolic complexity and accuracy of current GEMs constructed for other bacteria and, in addition to accounting for reduced metabolic coverage, it lacks the proper gene-protein associations (GPRs), cellular compartmentalization, and detailed biomass objective function (BOF). Therefore, the available GEM of *H. influenzae* cannot be used for complex systems biology studies and presents limited applicability (23). Here, we fill this gap by constructing and validating a high-quality GEM of *H. influenzae* strain RdKW20 (Fig. 1). This new model, *i*NL638, was used to screen therapeutic targets in this bacterium by means of gene essentiality analysis, identifying fatty acid metabolism as a promising one. We focused on the FabH $\beta$-ketoacyl-acyl carrier protein, i.e. ketoacyl-ACP synthase at the initiating step of fatty acid synthesis. The antimicrobial activity of a good FabH inhibitor was thoroughly characterized *in vitro* and *in vivo* at the preclinical level using a wide set of NTHi clinical strains.

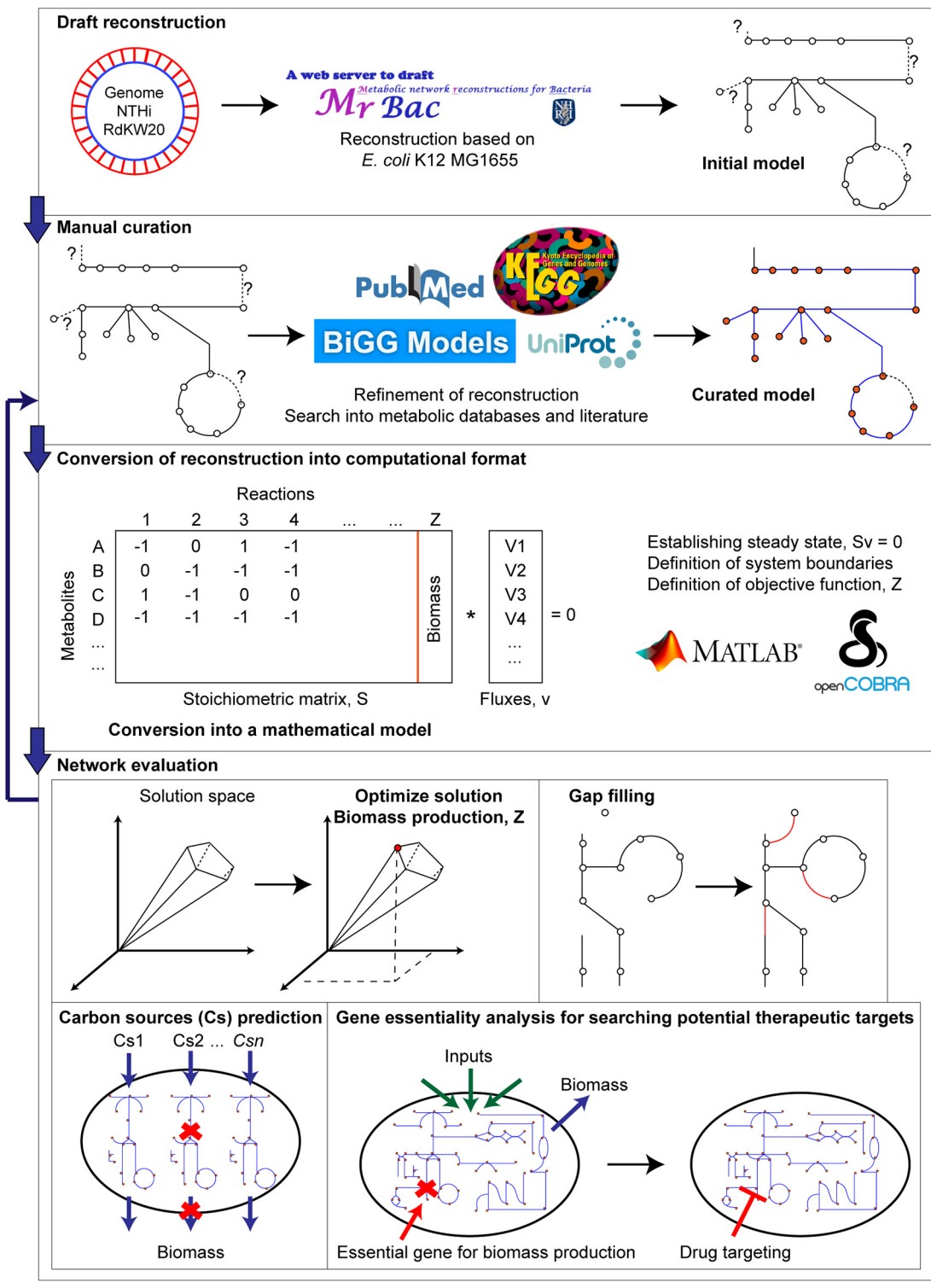

**FIG 1** Roadmap for using *H. influenzae* metabolic reconstruction as a screening platform.

## RESULTS

**Construction, main features, and consistency of *i*NL638, a high-quality model of *H. influenzae* RdKW20.** *H. influenzae* RdKW20 metabolic reconstruction was addressed by using a well-known 4-step protocol (24) and the last genomic annotation of this strain at NCBI (Fig. 1). A detailed description of model construction and manual curation is presented in Materials and Methods. The final model, *i*NL638, contains 638 genes, 1,385 reactions, and

**TABLE 1** Metabolic content of *i*NL638 compared with its antecessors

| Content | Value for: | | |
| --- | --- | --- | --- |
| | *i*NL638 (this study) | *i*CS400 (22) | *i*JE296 (19) |
| Metabolites | 1,161 | 451 | 343 |
| Unique | 746 | 367 | NA[a] |
| Cytoplasmic | 706 | 367 | NA |
| Periplasmic | 250 | 0 | NA |
| Extracellular | 205 | 84 | NA |
| Reactions | 1,385 | 546 | 488 |
| Metabolic | 786 | 374 | NA |
| Transport | 395 | 87 | NA |
| Exchange | 203 | 84 | NA |
| Orphan | 68 | 10 | NA |
| Blocked | 234 | 71 | NA |
| Genes | 638 | 400 | 296 |

[a]NA, data not available.

1,161 nonunique metabolites distributed into three cellular compartments, external, periplasm, and cytosol (Table 1). Compared to the previous *H. influenzae* model, *i*NL638 contains a significantly larger number of reactions and metabolites (Table 1). Reactions were classified into 12 major categories, of which transport and cell envelope biosynthesis are the largest groups, with 395 and 142, respectively (Fig. 2A). *i*NL638 was evaluated using the Memote tool (25) to define its completeness, consistency, and interoperability as a model while analyzing potential flaws or shortcomings. The model's overall score was 91%, which suggests very good completeness. A limitation was the lack of annotation to outside references for some genes, metabolites, and reactions. This will only have an affect when using the model with certain automated tools or scripts, but its accuracy or usability should not be affected. The model scored 99% for the critical category of consistency, which represents accuracy in stoichiometry, mass balances, charge balances, connectivity of metabolites, and reaction cycles. The Memote analysis demonstrated that *i*NL638 is a highly complete and detailed model that can be used as a reference for other GEM constructions. The model, scripts used in this study, and Memote full report are freely accessible through github (https://github.com/SBGlab/Haemophilus_influenzae_GEM).

***i*NL638 accurately predicts *H. influenzae* RdKW20 main metabolic features.** Large completeness and consistence does not always equate to a high-quality model, and GEMs need to be validated by assessing their ability to compute physiological states (24). To validate *i*NL638, we evaluated its capability of predicting well-known metabolic traits of RdKW20, such as the production of acetate and formate (26, 27). In the absence of available experimental carbon flux distribution data, we addressed the ability of *i*NL638 to predict metabolic end products under changing oxygen tensions. For this, we used CDM *in silico* media and monitored the excretion rates of acetate, formate, and hypoxanthine as a function of oxygen uptake. We identified three different steps (Fig. 2B). In the absence of oxygen, we computed formate and acetate as main end products driving the lower growth rate. Under these conditions, formate was produced by the action of the pyruvate formate lyase (PFL), while acetate was produced from acetyl-CoA by the combined action of phosphate acetyltransferase (PTA) and acetate kinase (ACK). *i*NL638 predicted PFL as a major source of acetyl-CoA, whereas the pyruvate dehydrogenase (PDH) only played a limited role (Fig. 2B). Suboptimal oxygen uptake rates led a second phase characterized by an oxygen-dependent increase of the growth rate. During this phase, the flux through PFL decreased linearly; in fact, no flux was predicted at oxygen uptakes higher than 7 mmol per gram of dry weight (gDW)·h$^{-1}$. As a consequence, formate secretion rates decreased in parallel. The role of acetyl-coenzyme A (CoA) as a main source was taken by the PDH, which completely replaced PFL at oxygen uptakes higher than 7 mmol/gDW·h$^{-1}$. Finally, under optimal oxygen availability, we observed a stable growth rate with acetate as the sole end product. These predictions agree with the switch in pyruvate metabolism from PFL to PDH, observed in *H.*

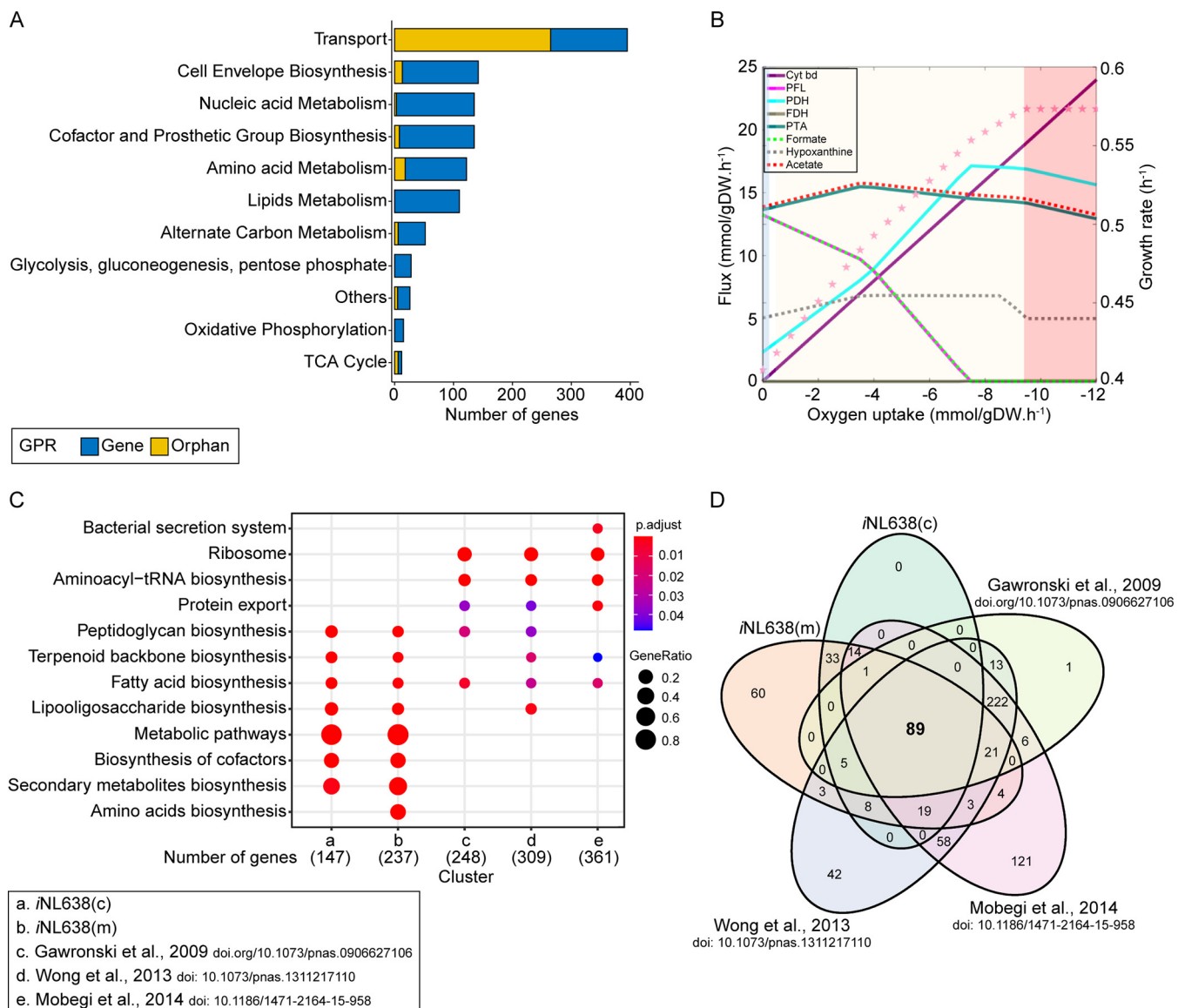

**FIG 2** Analysis of content and performance of *i*NL638. (A) Metabolic subsystem distribution of reactions of *i*NL638 model. Gene-associated reactions (blue) and non-gene-associated (orphan) reactions (yellow) are indicated. (B) Evolution of pyruvate metabolism and metabolic end product as a function of oxygen availability. The metabolic behavior of *H. influenzae* was predicted under anaerobic (blue), microaerobic (yellow), and aerobic (red) conditions. CDM using glucose and pyruvate as the carbon source was used. Fluxes of pyruvate dehydrogenase (PDH), pyruvate formate lyase (PFL), formate dehydrogenase (FDH), and phosphate acetyltransferase (PTA) were monitored as well as the excretion rates of formate, acetate, and hypoxanthine. (C) KEGG-based gene essential enrichment. Only genes annotated in KEGG were used for this analysis. We found 89 genes predicted to be essential among studies, mostly involved in the metabolism of fatty acids. (D) Venn diagram showing the intersections between predicted essential genes commonly found when comparing our *i*NL638 model-based screening and three previous independent studies.iNL638(c), gene essential enrichment when using complete CDM; iNLC638, gene essential enrichment when using CDM with a minimized composition, i.e minimal CDM or mCDM.iNL638(c), gene essential enrichment when using complete CDM; iNLC638, gene essential enrichment when using CDM with a minimized composition, i.e minimal CDM or mCDM.iNL638(c), gene essential enrichment when using complete CDM; iNLC638, gene essential enrichment when using CDM with a minimized composition, i.e minimal CDM or mCDM.iNL638(c), gene essential enrichment when using complete CDM; iNLC638, gene essential enrichment when using CDM with a minimized composition, i.e minimal CDM or mCDM.

*influenzae* as oxygen becomes available (27). Higher levels of NADH provided by PDH are further funneled to cytochrome *bd* oxidase via NADH:quinone reductase, driving the generation of energy, a behavior also accurately predicted by the model (Fig. 2B). No flux through formate dehydrogenase (FDH) was predicted under any condition, while only a slightly lower hypoxanthine secretion rate, resulting from the degradation of inosine, was predicted. These predictions are consistent with previous experimental observations (26).

Overall, *i*NL638 predicts main carbon metabolism and energy generation in *H. influenzae* with high accuracy even under variable oxygen availability. It is ready to be used as a powerful predictive tool.

**Defining condition-independent gene essentiality in *H. influenzae*.** The validated model was used to find metabolic drug targets based on gene essentiality predictions. Since gene essentiality assignment is largely dependent on culture conditions (28, 29), we evaluated this property in well-defined situations. We first addressed RdKW20 minimal nutrient requirements in CDM and defined a minimal medium only including those nutrients required for growth. Despite the presence of amino acids in the CDM composition, *i*NL638 grew in amino acid absence using nitrate as the nitrogen source, suggesting the lack of amino acid auxotrophies. Interestingly, utilization of nitrate as nitrogen source has been reported in *H. influenzae* (30–32). Similarly, *i*NL638 predicted as essential only a few additional CDM nutrients, including uracil, NAD, protoheme, choline, pantothenate, and thiamine. Since a robust drug target requires being condition independent, we evaluated the essentiality of *i*NL638 in this minimal CDM (mCDM) and in the complete formulation (cCDM). We found 260 and 169 essential genes using minimal and complete CDM, respectively (see Data Set S2 in the supplemental material). This analysis identified a total of 91 conditional essential genes whose requirement for growth is only visible using the mCDM formulation. We further evaluated the accuracy of *i*NL638 predictions against three available experimental gene essentiality studies (33–35). Overall, we found higher accuracies (0.72 to 0.761) and specificities (0.84 to 0.882) in predictions performed using cCDM (Fig. S1A). Experimental data sets were constructed using the rich medium supplemented brain heart infusion (sBHI). Therefore, most discrepancies between *in vivo* and *in silico* data are likely due to the different medium compositions. We found similar accuracy and sensitivities when comparing the different experimental studies to each other, suggesting that gene essentiality analysis is also study dependent (Fig. S1B).

The large diversity of gene essentiality data sets analyzed and discrepancies found led us to perform an enrichment analysis for both the *in silico* and experimental data sets to identify subsystems overrepresented in all cases (Fig. 2C). This analysis showed a significant enrichment of metabolic genes in the *in silico* data sets, while nonmetabolic genes were more abundant in experimental data sets. This is due to the fact that genes other than those encoding metabolic functions are not included in GEMs. Genes related to fatty acid metabolism were present in all available studies, suggesting this subsystem is a condition-independent essential metabolic hub. Indeed, from the 89 genes predicted to be essential under all conditions, 29 were involved in lipid metabolism (Fig. 2D, Table 2, and Data Set S2). Together, the gene essentiality analysis performed using *in silico* predictions with *i*NL638 in the context of previous experimental studies largely shows lipid metabolism has promising *H. influenzae* drug targets.

**Model-based search of *H. influenzae* essential genes highlights lipid metabolism targets.** Within lipid metabolism, our analysis predicted essentiality for 11 genes encoding fatty acid biosynthesis enzymes (FASII pathway), 6 genes encoding enzymes required for phospholipid biosynthesis, and 12 genes encoding enzymes involved in lipid A biosynthesis (Table 2 and Fig. S2 and Data Set S2). The *H. influenzae* FASII pathway generates the acyl-ACP (acyl carrier protein) and $\beta$-hydroxyacyl-ACP products, which are key components of the bacterial membrane. Acyl-ACP is used by the 1-acyl-*sn*-glycerol-3-phosphate (PlsB) and 1-acyl-*sn*-glycerol-3-phosphate (PlsC) acyltransferases to generate phosphatidic acid, the precursor of *H. influenzae* phospholipids phosphatidyl ethanolamine (PE) and phosphatidyl glycerol (PG) (36); $\beta$-hydroxyacyl-ACP molecules are substrates for the acyltransferases catalyzing the initial steps in the lipid A biosynthesis (37). Predicted essentiality and the selective targeting of the FASII pathway, due to significant differences in the structure of eukaryotic and bacterial fatty acid synthesis systems, make it an attractive target for drug discovery.

A key FASII enzyme is the $\beta$-ketoacyl-acyl carrier protein synthase III FabH, which initiates fatty acid elongation cycles by catalyzing the condensation reaction between a CoA-attached acyl group and an ACP-attached malonyl group, yielding $\beta$-ketoacyl-ACP. This gene was found to be essential both in computational and experimental analysis (Table 2 and Data Set S2). Moreover, we unsuccessfully tried to inactivate it in the RdKW20 strain (data not shown), largely supporting its essentiality under a variety of laboratory conditions. FabH has no significant homologous proteins in humans, and small-molecule inhibitors could lead to selective nontoxic antibacterials (38–40). Natural products and chemically synthesized FabH inhibitors

**TABLE 2** Genes predicted to be essential by *i*NL638-screening commonly found in references 33 to 35

| Pathway | Category | ED no. | Gene | Enzyme |
|---|---|---|---|---|
| Lipid metabolism | FASII | | HI0154 (*acpP*) | Acyl carrier protein |
| | | 1.1.1.100 | HI0155 (*fabG*) | 3-Oxoacyl-[acyl-carrier-protein] reductase FabG |
| | | 2.3.1.39 | HI0156 (*fabD*) | Malonyl CoA-acyl carrier protein transacylase |
| | | 2.3.1.180 | HI0157 (*fabH*) | 3-Oxoacyl-[acyl-carrier-protein] synthase 3 |
| | | 2.1.3.15 | HI0406 (*accA*) | Acetyl-coenzyme A carboxylase carboxyl transferase subunit alpha |
| | | | HI0971 (*accB*) | Biotin carboxyl carrier protein of acetyl-CoA carboxylase |
| | | 6.3.4.14, 6.4.1.2 | HI0972 (*accC*) | Biotin carboxylase |
| | | 2.1.3.15 | HI1260 (*accD*) | Acetyl-coenzyme A carboxylase carboxyl transferase subunit beta |
| | | 4.2.1.59, 5.3.3.14 | HI1325 (*fabA*) | 3-Hydroxydecanoyl-[acyl-carrier-protein] dehydratase |
| | | 2.3.1.41 | HI1533 (*fabB*) | 3-Oxoacyl-[acyl-carrier-protein] synthase 1 |
| | | 1.3.1.9 | HI1734 (*fabI*) | Enoyl-[acyl-carrier-protein] reductase [NADH] FabI |
| | Lipid A | 2.7.7.38 | HI0058 (*kdsB*) | 3-Deoxy-manno-octulosonate cytidylyltransferase |
| | | 2.7.1.130 | HI0059 (*lpxK*) | Tetraacyldisaccharide 4′-kinase |
| | | 7.5.2.6 | HI0060 (*msbA*) | ATP-dependent lipid A-core flippase |
| | | 2.7.1.166 | HI0260.1 (*kdkA*) | 3-Deoxy-D-manno-octulosonic acid kinase |
| | | 2.4.99.12 | HI0652 (*kdtA*) | 3-Deoxy-D-manno-octulosonic acid transferase |
| | | 3.6.1.54 | HI0735 (*lpxH*) | UDP-2,3-diacylglucosamine hydrolase |
| | | 2.3.1.- | HI0915 (*lpxD*) | UDP-3-O-acylglucosamine N-acyltransferase |
| | | 2.4.1.182 | HI1060 (*lpxB*) | Lipid-A-disaccharide synthase |
| | | 2.3.1.129 | HI1061 (*lpxA*) | Acyl-[acyl-carrier-protein]-UDP-N-acetylglucosamine O-acyltransferase |
| | | 3.5.1.108 | HI1144 (*lpxC*) | UDP-3-O-acyl-N-acetylglucosamine deacetylase |
| | | 2.3.1.241 | HI1527 (*htrB*) | Lipid A biosynthesis lauroyltransferase |
| | | 2.5.1.55 | HI1557 (*kdsA*) | 2-Dehydro-3-deoxyphosphooctonate aldolase |
| | Phospholipids | 2.7.8.5 | HI0123 (*pgsA*) | CDP-diacylglycerol-glycerol-3-phosphate 3-phosphatidyltransferase |
| | | 4.1.1.65 | HI0160 (*psd*) | Phosphatidylserine decarboxylase proenzyme |
| | | 2.7.1.107 | HI0335 (*dgkA*) | Diacylglycerol kinase |
| | | 2.7.8.8 | HI0425 (*pssA*) | CDP-diacylglycerol-serine O-phosphatidyltransferase |
| | | 2.3.1.15 | HI0748 (*plsB*) | Glycerol-3-phosphate acyltransferase |
| | | 2.7.7.41 | HI0919 (*cdsA*) | Phosphatidate cytidylyltransferase |
| Peptidoglycan biosynthesis | | 2.7.2.4, 1.1.1.3 | HI0089 (*thrA*) | Bifunctional aspartokinase/homoserine dehydrogenase |
| | | 3.5.1.18 | HI0102 (*dapE*) | Succinyl-diaminopimelate desuccinylase |
| | | 4.3.3.7 | HI0255 (*dapA*) | 4-Hydroxy-tetrahydrodipicolinate synthase |
| | | 1.3.1.98 | HI0268 (*murB*) | UDP-N-acetylenolpyruvoylglucosamine reductase |
| | | 2.4.1.129, 3.4.16.4 | HI0440 (*ponA*) | Penicillin-binding protein 1A |
| | | 2.7.7.23, 2.3.1.157 | HI0642 (*glmU*) | Bifunctional protein GlmU |
| | | 1.2.1.11 | HI0646 (*asd*) | Aspartate-semialdehyde dehydrogenase |
| | | 5.1.1.7 | HI0750 (*dapF*) | Diaminopimelate epimerase |
| | | 2.5.1.31 | HI0920 (*uppS*) | Di-trans,poly-cis-undecaprenyl-diphosphate synthase [(2E,6E)-farnesyl-diphosphate specific] |
| | | 2.5.1.7 | HI1081 (*murA*) | UDP-N-acetylglucosamine 1-carboxyvinyltransferase |
| | | 6.3.2.13 | HI1133 (*murE*) | UDP-N-acetylmuramoyl-L-alanyl-D-glutamate-2,6-diaminopimelate ligase |
| | | 6.3.2.10 | HI1134 (*murF*) | UDP-N-acetylmuramoyl-tripeptide-D-alanyl-D-alanine ligase |
| | | 2.7.8.13 | HI1135 (*mraY*) | Phospho-N-acetylmuramoyl-pentapeptide-transferase |
| | | 6.3.2.9 | HI1136 (*murD*) | UDP-N-acetylmuramoylalanine-D-glutamate ligase |
| | | 2.4.1.227 | HI1138 (*murG*) | UDP-N-acetylglucosamine-N-acetylmuramyl-(pentapeptide) pyrophosphoryl-undecaprenol N-acetylglucosamine transferase |
| | | 6.3.2.8 | HI1139 (*murC*) | UDP-N-acetylmuramate-L-alanine ligase |
| | | 6.3.2.4 | HI1140 (*ddl*) | D-Alanine-D-alanine ligase |
| | | 1.17.1.8 | HI1308 (*dapB*) | 4-hydroxy-tetrahydrodipicolinate reductase |
| | | 2.3.1.117 | HI1634 (*dapD*) | 2,3,4,5-tetrahydropyridine-2,6-dicarboxylate N-succinyltransferase |
| | | 5.1.1.3 | HI1739.2 (*murI*) | Glutamate racemase |
| Amino acids metabolism | Phenylalanine, tyrosine, and tryptophan | 4.2.3.5 | HI0196 (*aroC*) | Chorismate synthase |
| | | 2.7.1.71 | HI0207 (*aroK*) | Shikimate kinase |
| | | 4.2.3.4 | HI0208 (*aroB*) | 3-Dehydroquinate synthase |

**TABLE 2** (Continued)

| Pathway | Category | ED no. | Gene | Enzyme |
|---|---|---|---|---|
| | | 2.5.1.54 | HI1547 (*aroG*) | Phospho-2-dehydro-3-deoxyheptonate aldolase |
| | | 2.5.1.19 | HI1589 (*aroA*) | 3-Phosphoshikimate 1-carboxyvinyltransferase |
| | Methionine | 2.5.1.6 | HI1172 (*metK*) | *S*-Adenosylmethionine synthase |
| | | 3.2.2.9 | HI1216 (*mtnN*) | 5′-Methylthioadenosine/*S*-adenosylhomocysteine nucleosidase |
| | Alanine | 5.1.1.1 | HI1575 (*alr*) | Alanine racemase |
| Isopentenyl biosynthesis | | 1.17.7.3 | HI0368 (*ispG*) | 4-Hydroxy-3-methylbut-2-en-1-yl diphosphate synthase (flavodoxin) |
| | | 4.6.1.12 | HI0671 (*ispF*) | 2-C-methyl-D-erythritol 2,4-cyclodiphosphate synthase |
| | | 1.1.1.267 | HI0807 (*ispC, dxr*) | 1-Deoxy-D-xylulose 5-phosphate reductoisomerase |
| | | 2.5.1.90 | HI0881 (*ispB*) | Octaprenyl diphosphate synthase |
| | | 1.17.7.4 | HI1007 (*ispH*) | 4-Hydroxy-3-methylbut-2-enyl diphosphate reductase |
| | | 2.2.1.7 | HI1439 (*dxs*) | 1-Deoxy-D-xylulose-5-phosphate synthase |
| | | 2.7.1.148 | HI1608 (*ispE*) | 4-Diphosphocytidyl-2-C-methyl-D-erythritol kinase |
| Quinone biosynthesis | | 2.2.1.9 | HI0283 (*menD*) | 2-Succinyl-5-enolpyruvyl-6-hydroxy-3-cyclohexene-1-carboxylate synthase |
| | | 5.4.4.2 | HI0285 (*menF*) | Isochorismate synthase MenF |
| | | 2.5.1.74 | HI0509 (*menA*) | 1,4-Dihydroxy-2-naphthoate octaprenyltransferase |
| | | 4.1.3.36 | HI0968 (*menB*) | 1,4-Dihydroxy-2-naphthoyl-CoA synthase |
| | | 4.2.1.113 | HI0969 (*menC*) | o-Succinylbenzoate synthase |
| Transport systems | | | HI0139 (*ompP2*) | Outer membrane protein P2 |
| | | | HI0625 (*trkA*) | Trk system potassium uptake protein TrkA |
| | | | HI1035 (*corA*) | Magnesium transport protein CorA |
| | | | HI1641 (*sapD*) | Peptide transport system ATP-binding protein SapD |
| Nucleotide metabolism | | 2.7.4.3 | HI0349 (*adk*) | Adenylate kinase |
| | | 2.7.4.22 | HI1065 (*pyrH*) | Uridylate kinase |
| | | 2.7.4.8 | HI1743 (*gmk*) | Guanylate kinase |
| Iron-sulfur cluster metabolism | | | HI0376 (*iscA*) | Iron-binding protein IscA |
| | | | HI0377 (*iscU*) | Iron-sulfur cluster assembly scaffold protein IscU |
| | | 2.8.1.7 | HI0378 (*nifS*) | Cysteine desulfurase IscS |
| Vitamin metabolism | | 1.5.1.5, 3.5.4.9 | HI0609 (*folD*) | Bifunctional protein FolD |
| | | 1.5.1.3 | HI0899 (*folA*) | Dihydrofolate reductase |
| | | 2.7.1.26, 2.7.7.2 | HI0963 (*ribF*) | Bifunctional riboflavin kinase/FMN adenylyltransferase |
| Coenzyme A biosynthesis | | 2.7.1.33 | HI0631 (*coaA*) | Pantothenate kinase |
| | | 2.7.1.24 | HI0890m (*coaE*) | Dephospho-CoA kinase |
| | | 4.1.1.36, 6.3.2.5 | HI0953 (*coaBC*) | Coenzyme A biosynthesis bifunctional protein CoaBC |
| Protein modification | | 2.8.1.8 | HI0026 (*lipA*) | Lipoyl synthase |
| | | 2.3.1.181 | HI0027 (*lipB*) | Octanoyltransferase |
| NAD metabolism | | 2.7.1.23 | HI0072 (*nadK*) | NAD kinase |
| Sugar metabolism | | 2.2.1.1 | HI1023 (*tktA*) | Transketolase |

have been reported and half-maximal inhibitory concentrations ($IC_{50}$) determined for some of them (38, 41). The search for novel FabH inhibitors focused our attention on 1-(5-(2-fluoro-5-(hydroxymethyl)phenyl)pyridin-2-yl)piperidine-4-acetic acid (compound 31 in reference 38; here called FabHi) (Fig. 3B, left). This compound exhibited a potent inhibition of *H. influenzae* FabH activity ($IC_{50}$ of 0.82 $\mu$M), high solubility, acceptable human plasma protein binding, easy chemical accessibility (38), feasibility of visualizing its mode of binding by molecular docking, and, therefore, the possibility of improving favorable contacts through rational design.

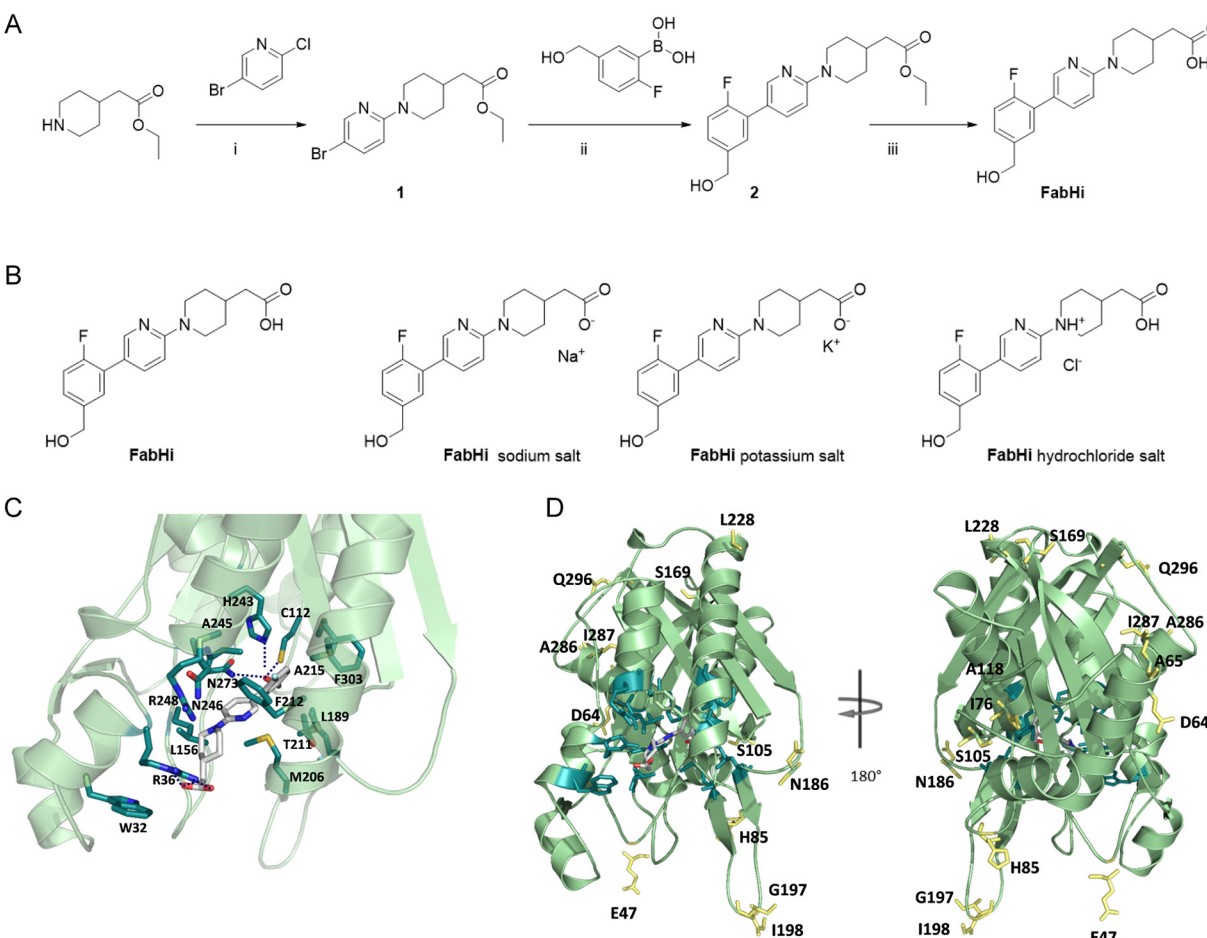

**FIG 3** FabHi synthesis, structure, and mode of binding. (A) Synthesis of FabHi. Reagents and conditions were (i) triethylamine (TEA) TEA, anhydrous acetonitrile, microwave $M_w$ 125°C, 2 h; (ii) Pd(ddpf)Cl$_2$, K$_2$CO$_3$, anhydrous DMF, 75°C, 16 h; (iii) 1 M NaOH, THF-MeOH (1:1), reflux, 4 h. (B) Structure of FabHi and its sodium, potassium, and hydrochloride salts. (C) Stick representation of FabHi in the best complex model with FabH$_{RdKW20}$ obtained with AutoDock4.2. FabH (cartoon) is depicted in light green with side chains of residues involved in inhibitor binding in stick representation (dark green); polar contacts are represented as dotted lines. Some relevant residues are key in CoA binding (Trp32, Arg36, Phe212, and Asn246) and likely in ACP binding (Arg36 and Arg248) according to *E. coli* FabH structures (52). (D) Residues showing polymorphism in FabH allelic variants, depicted as yellow sticks, locate outside de substrate-binding pocket (FabHi binding-residues in dark green).

***In silico* envisioning of FabHi binding mode.** Despite FabHi having promising activity, its action mode is completely unknown, so we first addressed its mode of action by molecular docking. FabH displays two similarly folded N-terminal and C-terminal halves, which differ in loops and insertion sequences (Fig. S3) (42). FabH core primarily provides a supportive structural scaffold, while insertion sequences are mainly involved in substrate recognition and catalysis. To visualize the interaction mode, FabHi was docked into the active site of FabH$_{RdKW20}$. The lowest-energy complex, obtained with AutoDock, placed FabHi at the substrate binding cavity with the hydroxymethyl group of the phenyl ring hydrogen bonded to the Cys112–His243–Asn273 catalytic triad and the carboxylic group of the piperidine ring forming three additional hydrogen bonds with Arg36 (Fig. 3C and Fig. S4A to D). The 2-fluoro-5-(hydroxymethyl)phenyl ring also makes many favorable hydrophobic interactions through the ring carbon atoms (Met206, Thr211, Ala215, Ala245, and Phe303), the hydroxylmethyl chain (Cys112, Leu189, His243, and Asn273), and the fluoro group (Ala215), as depicted in Fig. S4C. Hydrophobic contacts with Ala245 further extend to the pyridine ring, whose nitrogen atom interacts with Phe212. The piperidine ring makes additional contacts through its carbon atoms (Asn246, Arg248, Leu156, and Arg36) and the carboxylic group (Arg36). Van der Waals interactions might also contribute to stabilize the complex. The intricate network of inhibitor-protein interactions predicted for the FabHi:FabH$_{RdKW20}$ complex fairly agrees with the contacts described for structurally related inhibitors in complex with *Escherichia coli* FabH (38), as does the inhibitor disposition within the binding cavity (Fig. S4D). Single point mutations at

**TABLE 3** Distribution of FabH variation across a previously whole-genome sequenced (WGS) collection of NTHi clinical isolates

| FabH variant | No. of strains | Frequency (%) | NTHi WGS clinical strain(s) | Selected representative strain |
|---|---|---|---|---|
| A1 | 29 | 30.9 | P667, P668, P669, P594, P595, P596, P650, P676, P679, P853, P627, P628, P671, P610, P617, P634, P635, P636, P637, P661, P597, P639, P662, P663, P598, P631, P611, P606, NTHi375 | NTHi375[b] |
| A2 | 17 | 18.1 | P600, P601, P602, P612, P613, P614, P615, P616, P618, P620, P621, P622, P623, P624, P629, P632, P633 | P621 |
| A3 | 12 | 12.8 | P670, P672, P674, P675, P677, P678, P646, P647, P648, P649, P592, RdKW20 | RdKW20[a] |
| A4 | 6 | 6.4 | P609, P599, P651, P652, P653, P654 | P652 |
| A5 | 6 | 6.4 | P619, P640, P638, P625, P673, P630 | P673 |
| A6 | 5 | 5.3 | P641, P642, P588, P604, P605 | P642 |
| A7 | 4 | 4.3 | P664, P665, P666, P591 | P665 |
| A8 | 3 | 3.2 | P657, P658, P660 | P657 |
| A9 | 3 | 3.2 | P643, P644, P645 | P645 |
| A10 | 2 | 2.1 | P603, P851 | P851 |
| A11 | 2 | 2.1 | P656, P593 | P593 |
| A12 | 2 | 2.1 | P607, P608 | P607 |
| A13 | 2 | 2.1 | P589, P590 | P590 |
| A14 | 1 | 1.1 | P626 | P626 |

[a]RdKW20 is a reference strain used to generate the *i*NL638 metabolic model.
[b]NTHi375 is an otitis media clinical isolate previously used in host-pathogen interplay studies (47–50).

positions predicted to interact with the inhibitor (Ala215Val, Ala215Ser, or Ala245Ser) significantly increased the resistance of *H. influenzae* to a close structural analogue of FabHi (38), supporting the plausibility of our docking model.

**FabHi has an antimicrobial effect on *H. influenzae* growth.** Before antimicrobial effect testing, we tried to improve FabHi solubility by preparing different salts (Fig. 3B). Assays were performed with the free acid form, as it produced a higher and more homogeneous loss of *H. influenzae* viability.

Our metabolic model and FabH$_{RdKW20}$:FabHi complex structure relied on the *H. influenzae* RdKW20 genome content and FabH$_{RdKW20}$ crystal structure (42, 43), respectively. However, *H. influenzae* strains show high genomic and phenotypic diversity (3, 44). To anticipate the potential role of FabHi as a broad inhibitor of *H. influenzae*, we examined the sequence variability of the *fabH* locus across a well-characterized genome-sequenced set of strains collected from COPD sputum samples over time, grouped in clonal types (CTs) (45, 46). Being present in all strains, best hits to *fabH* were extracted, followed by translation and multiple alignments. The NTHi375 reference strain was also included in this analysis (47–50). Fourteen different *fabH* allelic variants were found (named alleles 1 to 14, i.e., A1 to A14), with different frequency and CT distribution but intra-CT conservation (Table 3, Fig. S3 and data not shown). We selected one strain per FabH allelic variant to assess the antimicrobial activity of synthesized FabHi.

We first tested the susceptibility of the 14 selected *H. influenzae* strains to FabHi upon growth in CDM, being the medium used to interrogate the *i*NL638 model (strain growth shown in Fig. S5A). We observed a significant dose-dependent reduction of bacterial viability after incubation. Such an effect was heterogeneous among strains, suggesting that FabHi is a broad-range inhibitor of FabH$_{H.influenzae}$, irrespective of the allelic variant. Based on the slope of the survival curve and the extent of bacterial viability loss upon drug incubation (Fig. S5C to F), we established four strain categories, from higher to lower susceptibility: (i) strains showing a sharp slope reaching complete viability loss, like those carrying *fabH* variants A5-P673, A11-P593, and A13-P590 (blue); (ii) sharp slope without complete viability loss (A4-P652, A6-P642, A9-P645, and A10-P851 *fabH* variants; green); (iii) not-as-sharp slope but reaching complete viability loss (A2-P621, A7-P665, A12-P607, and A14-P626; red); and (iv) not-as-sharp slope, without complete viability loss (A1-NTHi375, A3-RdKW20, and A8-P657; orange).

CDM, useful to connect our computational and experimental approaches, is not used as a standard medium for *H. influenzae* growth and antimicrobial susceptibility testing (51). Thus, we tested susceptibility to FabHi of same strains grown in sBHI (growth shown in Fig. S5B). We observed, again, a significant and heterogeneous dose-dependent reduction of bacterial viability (Fig. 4A). Strain behavior was comparable to that in CDM in most

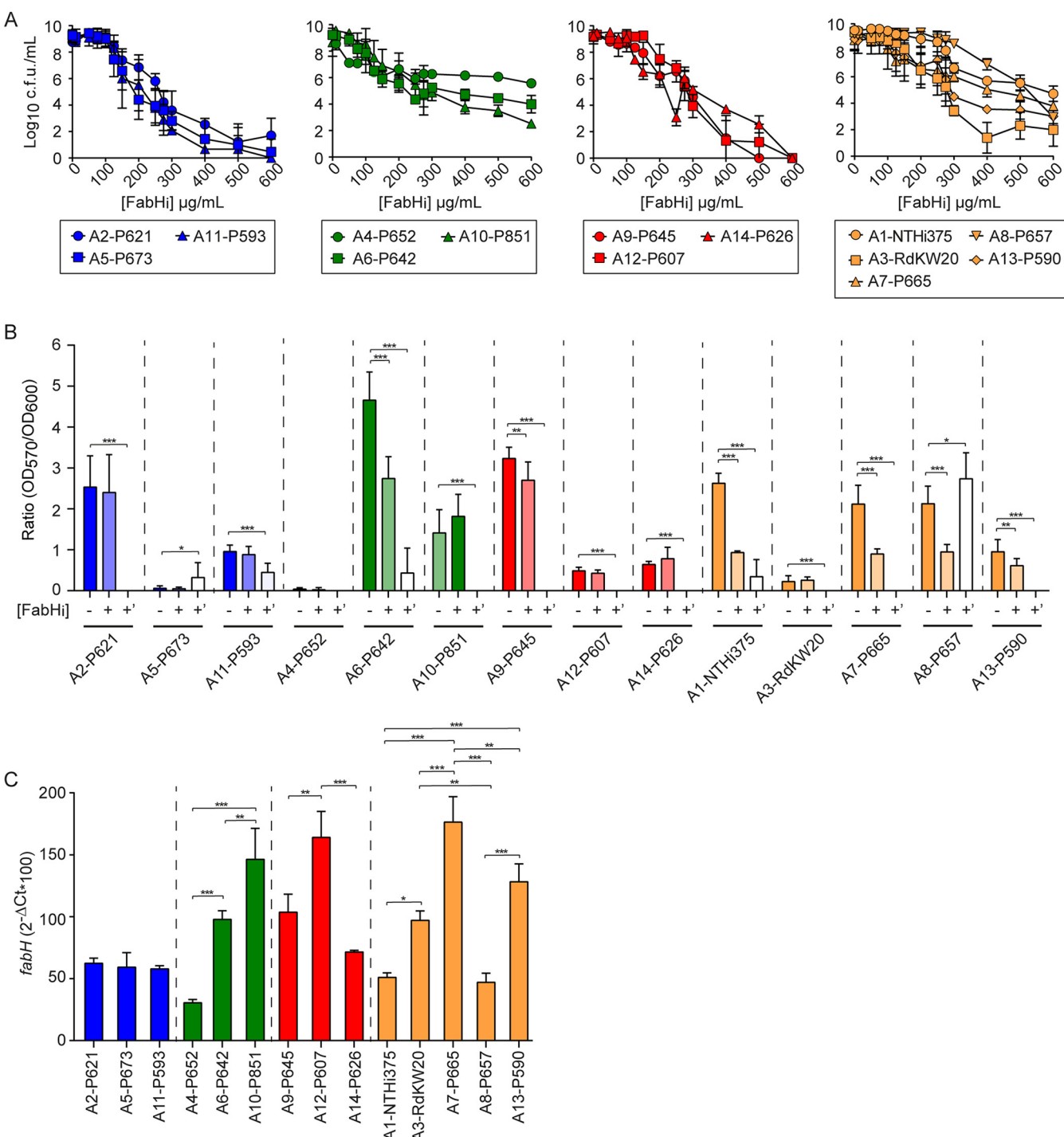

**FIG 4** FabHi has an antimicrobial effect on *H. influenzae* upon planktonic and biofilm growth. (A) Determination of FabHi inhibitory effect on representative NTHi strains carrying FabH variants A1 to A14 upon planktonic growth in sBHI medium. Susceptibility to FabHi was dose dependent. FabHi concentrations tested range from 10 to 600 $\mu$g/mL. Strains were classified into four groups (blue, green, red, and orange labeling) based on increasing resistance. Results are shown as $\log_{10}$ CFU/mL (mean $\pm$ SD). For each strain, statistical comparisons of means were performed by one-way ANOVA and Dunnett's multiple-comparison test. A reduction on bacterial viability was observed: A1-NTHi375, at [FabHi] 200 $\mu$g/mL and higher, $P < 0.0005$; A2-P621, at [FabHi] 200 $\mu$g/mL and higher, $P < 0.0001$; A3-RdKW20, at [FabHi] 150 $\mu$g/mL and higher, $P < 0.005$; A4-P652, at [FabHi] 50 $\mu$g/mL and higher, $P < 0.0001$; A5-P673, at [FabHi] 125 $\mu$g/mL and higher, $P < 0.0001$; A6-P642, at [FabHi] 75 $\mu$g/mL and higher, $P < 0.005$; A7-P665, at [FabHi] 125 $\mu$g/mL and higher, $P < 0.05$; A8-P657, at [FabH$_i$] 275 $\mu$g/mL and higher, $P < 0.05$; A9-P645, at [FabHi] 150 $\mu$g/mL and higher, $P < 0.05$; A10-P851, at [FabHi] 150 $\mu$g/mL and higher, $P < 0.005$; A11-P593, at [FabHi] 150 $\mu$g/mL and higher, $P < 0.0001$; A12-P607, at [FabHi] 200 $\mu$g/mL and higher, $P < 0.0001$; A13-P590, at [FabHi] 200 $\mu$g/mL and higher, $P < 0.05$; A14-P626, at [FabHi] 125 $\mu$g/mL and higher, $P < 0.0001$. (B) Determination of FabHi inhibitory effect on biofilm growth by representative NTHi strains carrying FabH variants A1 to A14. Strains were grown in the absence ($-$) or presence ($+$, $+'$) of FabHi. Two FabHi concentrations were used: (i) 50 $\mu$g/mL ($+$); (ii) a higher subinhibitory [FabHi] ($+'$) close to the minimum bactericidal concentration determined in panel A

cases, with some exceptions, and strain P590 carrying the A13 allelic variant was the clearest case (Fig. 4A and Fig. S5C). Detailed analysis of these curves suggests a biphasic mode with a fast initial reduction of bacterial viability, followed by a second phase where viability loss drops slower, in some cases up to undetectable levels. Changes related to the FabH variant and the growth medium mainly affect the bacterial response to FabHi concentrations in this second phase (Fig. S5H to K).

Likewise, we assessed the effect of FabHi on bacterial biofilm growth, first defining biofilm formation for each strain, which was heterogeneous from clear biofilm formers (A6-P642) to those forming poorly detectable biofilms (A4-P652 and A5-P673). FabHi 50 $\mu$g/mL (+), which allowed growth of all tested strains (Fig. 4A), lowered biofilm growth by A1-NTHi375, A6-P642, A7-P665, A8-P657, A9-P645, and A13-P590 strains (Fig. 4B, column +), but this inhibitory effect was not observed on strains belonging to the more susceptible category (for example, A2-P621 and A11-P593). A significant inhibitory effect was observed when using higher FabHi concentrations (Fig. 4B, column +').

In summary, the viability of *H. influenzae* clinical strains representative of the observed FabH variation decreased when exposed to FabHi in a dose-dependent manner and with variable efficacy among strains. The susceptibility of *H. influenzae* to FabHi irrespective of the culture conditions argues in favor of FabH as a condition-independent drug target and highlights our gene essentiality analysis as a robust computational approach to identify drug targets.

**Heterogeneity of FabHi antimicrobial effect among strains: assessing possible causes.** We next explored possible reasons accounting for the heterogeneity of FabHi antimicrobial effect. First, mapping of sequence polymorphisms in the FabH$_{RdKW20}$ structure showed that amino acid substitutions were out of the substrate-binding cavity (Fig. 3D). However, functional FabH is a dimer (42), and several of these polymorphisms located at the monomer-monomer interface could be relevant for dimer stabilization and even substrate selectivity, while others might affect the monomer conformation.

The loops comprising Ser84–Ser89 and Lys185–Gln207, with polymorphisms at positions 85, 186, 197, and 198, are involved in forming the dimer interface and, more importantly, in ligand binding or active-site formation (Fig. S4E). Thus, dimerization brings Tyr87 and adjacent residues close to the catalytic Cys112 of the dimeric partner (Fig. S4F), creating a complex network of contacts with residues from the other subunit (Thr81, Cys112, Val190, Leu191, Ala192, Gln193, Ile204, and Gly306) in which His85 itself participates. Residues at positions 197 and 198 conform the central part of the other loop turn whose pairing with the equivalent loop of the complementary subunit stabilizes the dimer interface, making contacts with Arg144, Ser201, Gly202, and Tyr203 from the other monomer (Fig. S4G). The loop is additionally involved in CoA recognition through Leu189 and Met206 moieties (52). Both actions could be sensitive to the polymorphism found at position 186 as well due to its proximity to the dimer interface (Fig. S4H). Analogously, Ser105 of FabH$_{RdKW20}$, replaced by cysteine in four variants, interacts with residues from the other subunit (Fig. S4H), and Ala118 (replaced by valine in one allelic variant) is close to the dimer interface. Polymorphisms potentially relevant for the monomer structural integrity comprise residue 76 (included in the hydrophobic core of the N-terminal domain), residue 169 (helping to maintain the relative dispositions of strands N$\beta$1 and N$\beta$5 of the N-terminal domain through hydrogen bonding to Asn2 and Asn171; Fig. S4I), and residues 286, 287, and 296 (orienting C$\beta$2, C$\alpha$3, C$\beta$3, and C$\beta$4 components of the C-terminal domain and helping to configure the loop connecting C$\alpha$3 to C$\beta$3; Fig. S4I). Finally, Ser228 (replaced by cysteine in three variants) is at the C terminus of C$\alpha$1, a long helix proposed to conform, together with C$\alpha$2, the binding surface for the ACP protein outside the active-site tunnel (52).

Therefore, sequence changes in any of these positions might have a significant impact

**FIG 4** Legend (Continued)

for each strain (275 $\mu$g/mL for A1-NTHI375, A3-RdKW20, A4-P652, A7-P665, A8-P657, and A14-P626; 200 $\mu$g/mL for A2-P621, A9-P645, A10-P851, A12-P607, and A13-P590; 100 $\mu$g/mL for A5-P673, A6-P642, and A11-P593). For each strain, statistical comparisons of means were performed by one-way ANOVA and Dunnett's multiple-comparison test (*, $P < 0.05$; **, $P < 0.005$; ***, $P \leq 0.0001$). (C) The *fabH* gene expression on representative NTHi strains used for inhibition studies, grown in sBHI to half of each strain's maximal OD$_{600}$. Statistical comparisons of means were performed for each previously established group by one-way ANOVA and Tukey's multiple-comparison test (*, $P < 0.05$; **, $P < 0.005$; ***, $P < 0.0001$).

on the functioning of the FabH enzyme, including the affinities for substrates, feedback inhibitor, and FabHi, as well as the catalytic efficiency and, therefore, FabH inhibitory capacity.

Second, we asked if the variable effect of FabHi would relate also to differences in the *fabH* gene expression. However, gene expression was heterogeneous, and we did not observe either a correlation between resistance and *fabH* gene expression, or a defined gene expression pattern for strains belonging to the previously established categories (Fig. 4C and Fig. S5G).

Third, previous evaluation of the FabHi analogue 1-(5-(2,5-dimethyl-3-(hydroxymethyl)phenyl)pyridin-2-yl)piperidine-4-carboxylic acid showed increased susceptibility upon inactivation of the AcrAB-TolC efflux pump system (38). AcrAB-TolC is present in the set of 14 selected strains, displaying some polymorphisms, although the AcrB hydrophobic trap (Gly142 or Ile143, Phe182, Glu594, Met599, Ile601, and Ile613) is fully conserved (53 and data not shown). Finally, although compensatory effects have been described in *Staphylococcus aureus* by the means of *fabH* mutations conferring resistance to FabF-directed antibiotics (54), this is unlikely for *H. influenzae*, as the *fabF* gene is absent from RdKW20 (55) and the rest of the tested strains.

**FabHi antimicrobial effect does not exclusively relate to the *fabH* gene allelic variant.** To further assess if the observed heterogeneity would match only to *fabH* gene allelic variation, we tested strains carrying the same variant. Variants A1 (NTHi375 as reference), A3 (RdKW20 as reference), A5 (P673 as reference), A6 (P642 as reference), and A8 (P657 as reference) were used as representative examples (Fig. 5). No significant differences were observed for strains containing variant A8, i.e., strains P657, P658, and P660, all belonging to the same CT. However, heterogeneity was observed in strain groups containing the A1, A3, A5, and A6 variants grouping different CTs. A1-containing strains P597, P639, and P606 were more susceptible while P667 and P671 were more resistant to FabHi than NTHi375; A3-containing strains P648 and P649 were more resistant and P670, P675, and P678 more susceptible to FabHi than RdKW20; A5-containing strains P619, P640, P638, and P630 were more resistant to FabHi than P673; A6-containing strains P588 and P641 were more resistant, and P604 and P605 more susceptible to FabHi than P642. In conclusion, the extent of the FabHi antimicrobial effect was not exclusively related to the FabH variant carried by each strain. Instead, the metabolic context provided by each strain may contribute to tailoring FabHi inhibitory effects.

**FabHi does not induce bacterial resistance, antibiotic synergies, or cytotoxicity.** To expand our FabHi antimicrobial characterization, we assessed the ability of strain P642, employed for *in vivo* assays (see below), to become resistant to FabHi through serial independent passage in sBHI broth containing three different inhibitory concentrations of FabHi (56, 57). After 12 consecutive passages, no growth was detected (Fig. 6A). FabHi and the antibiotic azithromycin (Azm) or ampicillin (Amp) were combined at different proportions and tested against P642 by the checkerboard method; antagonism or synergy effects were not detected. Figure 6B shows the MIC $\pm$ standard deviations (SD) for each compound separately, the MIC $\pm$ SD of the combinations tested, calculated fractional inhibitory concentration existence of synergy ($\Sigma$FIC) $\pm$ SD, and their interpretation.

FabHi effect on bacterial viability was also titrated in Earle's balanced salts (EBSS) medium aiming to mimic cultured epithelial cell infection conditions, showing a significant dose-dependent decrease (Fig. 6C). Based on this observation, A549 cells were pretreated with EBSS containing FabHi for 16 h, which was then replaced by EBSS medium without drug prior to infection. Under these conditions, FabHi did not show cytotoxicity (Fig. S6A). Cell exposure to increasing doses of FabHi did not modify P642 epithelial adhesion and invasion rates (Fig. 6D and E). Therefore, FabHi does not seem to modulate NTHi infection of airway epithelial cells under the tested conditions.

**FabHi efficacy preclinical testing reveals a protective effect on zebrafish systemic infection with NTHi.** Our final goal was to evaluate drug efficacy at the preclinical level, on a FabHi protective assay in adult zebrafish infected with NTHi, by following a previously established sepsis model (57, 58). We used strains P673 and P642, belonging to more FabHi-susceptible categories, to assess their capacity to infect zebrafish by injection of 3 animal groups (*n* = 10) with 10 $\mu$L of bacterial suspensions containing $7 \times 10^{10}$ CFU/mL or 0.9% saline solution as a control. NTHi P642 severely

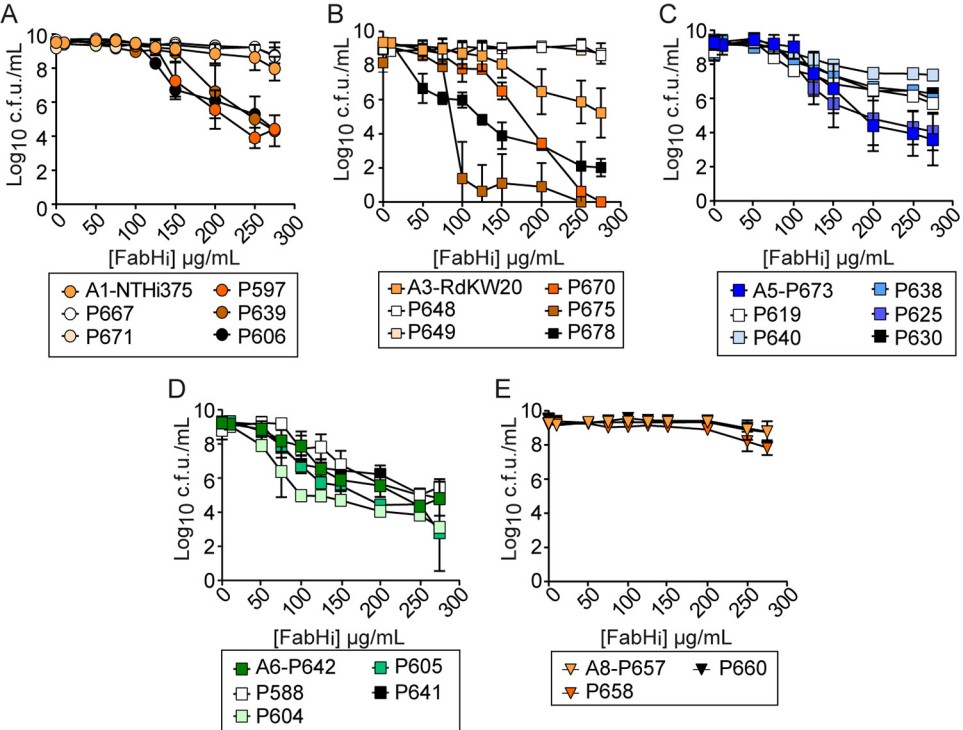

**FIG 5** FabHi effect on *H. influenzae* does not merely relate to *fabH* gene allelic variant. Determination of FabHi inhibitory effect on NTHi strains carrying FabH variants A1 (strains NTHi375, P667, P671, P597, P639, and P606) (A), A3 (strains RdKW20, P648, P649, P670, P675, and P678) (B), A5 (strains P673, P619, P640, P638, P625, and P630) (C), A6 (strains P642, P588, P604, P605, and P641) (D), or A8 (strains P657, P658, and P660) (E) upon planktonic growth in sBHi. Strains were susceptible to FabHi in a dose-dependent manner. FabHi concentrations testing in a range from 10 to 275 $\mu$g/mL. Results are shown as $\log_{10}$ CFU/mL (means ± SD). For each strain set containing the same FabH variant, statistical comparisons of means were performed by two-way analysis of variance (ANOVA) and Dunnett's multiple-comparisons test, using the previously used representative strain as a reference. A significant reduction on bacterial viability was observed. (A) A1 variant group, higher susceptibility than A1-NTHi375 reference strain is shown for P606 at [FabHi] > 125 $\mu$g/mL, $P < 0.005$; for P597 at [FabHi] > 150 $\mu$g/mL, $P < 0.0001$; for P639 at [FabHi] > 200 $\mu$g/mL, $P < 0.0001$; and lower susceptibility than the A1-NTHi375 reference strain is shown for P667 and P671 at [FabHi] > 275 $\mu$g/mL, $P < 0.05$. (B) A3 variant group, higher susceptibility than the A3-RdKW20 reference strain is shown for P670 at [FabHi] > 150 $\mu$g/mL, $P < 0.005$; for P675 at [FabHi] > 100 $\mu$g/mL, $P < 0.0001$; for P678 at [FabHi] > 50 $\mu$g/mL, $P < 0.0001$; and lower susceptibility than the A3-RdKW20 reference strain is shown for P648 at [FabHi] > 200 $\mu$g/mL, $P < 0.0001$; for P649 at [FabHi] > 200 $\mu$g/mL, $P < 0.0001$. (C) A5 variant group, lower susceptibility than A5-P673 reference strain is shown for P619 at [FabHi] 100 and >200 $\mu$g/mL, $P < 0.05$; for P640 at [FabHi] > 150 $\mu$g/mL, $P < 0.05$; for P638 and P630 at [FabHi] > 200 $\mu$g/mL, $P < 0.0001$. (D) A6 variant group, higher susceptibility than the A6-P642 reference strain is shown for P604 at [FabHi] > 75 $\mu$g/mL, $P < 0.05$; for P605 at [FabHi] > 125 $\mu$g/mL, $P < 0.005$; and lower susceptibility than the A6-P642 reference strain is shown for P588 at [FabHi] 100 and >150 $\mu$g/mL, $P < 0.05$; for P641 at [FabHi] 125 $\mu$g/mL, $P < 0.05$. (E) A8 variant group, no significant differences were found.

reduced zebrafish survival, and an infection dose consisting of $7 \times 10^8$ CFU/zebrafish caused progressive death of the animals after injection; however, strain P673 did not modify zebrafish survival (data not shown), precluding us from using this strain in this model system.

Next, a FabHi acute toxicity assay was performed in zebrafish embryos by following the fish embryo acute toxicity (FET) test. Results showed that FabHi doses lower than 5.7 $\mu$g/embryo were nontoxic (Fig. S6B). Based on these data, we tested FabHi's effect on P642-infected zebrafish by using a therapeutic regimen of intraperitoneal 4 $\mu$g FabHi per zebrafish, consisting of one administration 1 h postinfection (hpi). Survival for FabHi-treated and control untreated groups was monitored three times per day up to 4 days postinfection. The mortality rate in FabHi-treated infected zebrafish was significantly lower than in infected control animals receiving vehicle solution ($P < 0.0001$) (Fig. 6F). This model system rendered significantly FabHi-mediated increased survival upon zebrafish NTHi infection by using the P642 isolate.

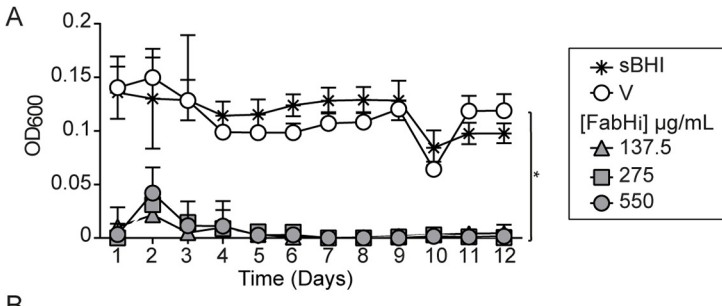

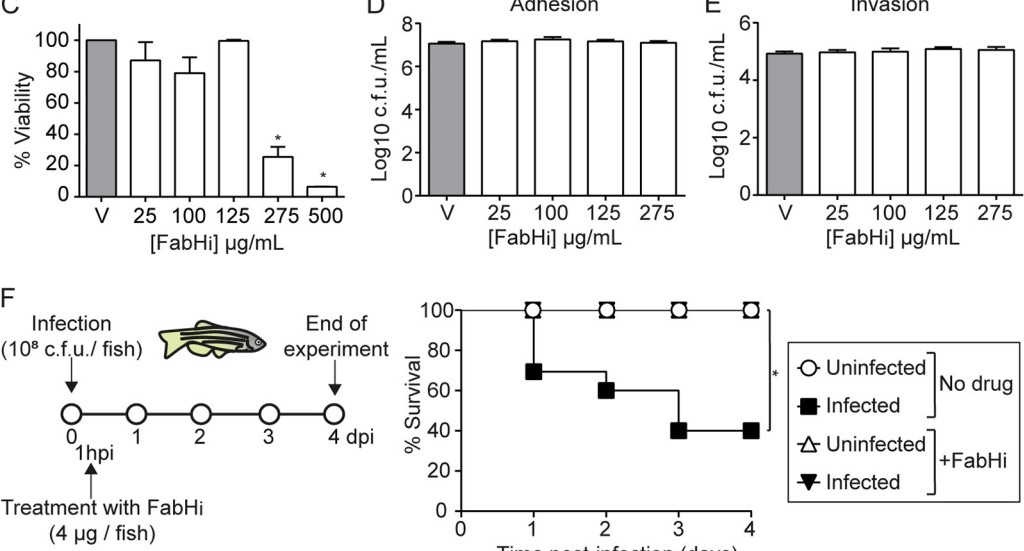

**FIG 6** FabHi *in vitro* characterization and preclinical evaluation. (A) A6-P642 did not grow after 12 daily serial passages in the presence of three [FabHi] (137.5, 275, and 550 μg/mL). Data are shown as $OD_{600}$ (means ± SD) in every passage (*, $P < 0.0001$). As controls, bacteria were grown in sBHI in the absence (asterisk) or presence of DMSO vehicle solution (V, white circle), with a volume corresponding to that used when testing FabHi at 550 μg/mL. Statistical comparisons of means were performed by one-way ANOVA and Dunnett's multiple-comparison test. (B) The checkboard method for strain P642, when combining FabHi-antibiotic molecules (means ± SD). (C) The viability of NTHi strain P642 was tested by simulating host cell infection conditions (2 h at 37°C in EBSS medium in the absence [V]/presence of FabHi). FabHi reduced (*, $P < 0.0001$) bacterial viability (means ± SD) in a dose-dependent manner. Based on these results, cell infection assays shown in panels D and E were performed by cell pretreatment with FabHi for 16 h and drug removal before infection. Controls (V) were cells that did not receive FabHi but did receive vehicle solution, i.e., DMSO. Adhesion (D) and invasion (E) assays did not render significant differences. Statistical comparisons of the means were performed with one-way ANOVA and Dunnett's multiple-comparison test. (F) Zebrafish were infected intraperitoneally with P642, $7 \times 10^8$ CFU/zebrafish. When necessary, FabHi at 4 μg/zebrafish was administered intraperitoneally at 1 hpi. Noninfected groups were administered PBS (white circle) or FabHi (white triangle); infected groups were administered PBS (black square) or FabHi (black inverted triangle). Survival rate is reported as percentage (means ± SD) of adult individual survival up to 4 days. Survival of NTHi-infected zebrafish was significantly higher in FabHi-treated than in untreated animals (*, $P < 0.0001$). To draw and analyze the Kaplan-Meier survival curve, a log-rank (Mantel-cox) test was used (*, $P < 0.0001$). Statistical comparisons between survival rates after 4 days were performed by one-way ANOVA and Sidak's multiple-comparison test.

## DISCUSSION

***i*NL638 is a powerful tool for drug target discovery and sets the bases of future *H. influenzae* strain-specific metabolic evaluation.** Here, we constructed a high-quality GEM of *H. influenzae* as a basic computational framework to identify essential genes (among

other possibilities) suitable for further exploitation as drug targets. The exponential increase of pathogen genomes and the emergence of high-throughput technologies are providing a myriad of large data sets. Among the numerous computational methods developed to generate meaningful outcomes of these data, metabolic modeling is emerging as a promising approach for the coherent organization of this information while providing mechanistic insights on biological systems (18). Thus, it is not surprising that this approach is being increasingly incorporated as a key strategy for antimicrobial target discovery at the level of preclinical research (16, 17, 59–63). Metabolic modeling has been used for the design of new drugs by informing target selection (16, 17, 64) and for the engineering of cells by rewiring metabolism toward the production of a product of interest (65, 66). Despite the success of GEMs, the lack of a high-quality model for *H. influenzae* largely hampered the application of such approaches in this clinically relevant bacterium. *i*NL638 represents a powerful and unprecedented computational tool for the search for novel drug targets in *H. influenzae*. In addition, the high quality of *i*NL638, in terms of consistency and interoperability, paves the way for the future multistrain modeling of the *H. influenzae* species using the strain RdKW20 as a reference. As shown by the strain-dependent effect of the inhibitor tested, the impact of metabolic context is gaining relevance as a key factor driving cell behavior (67). Therefore, further *H. influenzae* multistrain modeling effort could lead to the identification of secondary metabolic hubs modulating the response of known and novel drugs, promoting the establishment of synergistic and more robust antibacterial treatments. Following this notion, *Salmonella* multistrain modeling identified serovar-specific metabolic traits, including auxotrophies and catabolic pathways related to adaptations to their colonization sites (68), and strain-specific metabolic features were unraveled for *S. aureus*, *P. aeruginosa*, or *A. baumannii* (64, 69, 70).

**The potential of targeting *H. influenzae* fatty acid biosynthesis.** Among essential genes predicted by *i*NL638, 89 also were shown to be essential in three previous Tn-seq-based screenings (33–35). These data not only argue in favor of the accuracy of *i*NL638 when predicting essential genes but give further support to the potential role of these genes as drug targets. About 33% of predicted essential genes were involved in lipid metabolism. Although we focused here on FASII inhibition, we acknowledge that peptidoglycan biosynthesis, accounting for over 22% of the commonly shared predicted essential genes, also deserves attention, as the diversity of antimicrobials with distinct mechanisms reported to disrupt the bacterial peptidoglycan is wide (71). When focusing on FabH targeting, natural (41, 72–74) and synthetic molecules (38–41, 75, 76) have been reported. Here, we went further with compound 31, previously reported in reference 38, renamed FabHi for simplicity.

Several technical aspects of our FabHi antibacterial effect evaluation may deserve consideration by those interested in assessing novel drugs without preestablished standardized procedures. We initially aimed to adapt regular MIC assays by measuring culture turbidity, in combination with quantifying redox activity in metabolically active bacterial cells by reduction of iodonitrotetrazolium chloride (77) (data not shown), and with serial dilution plating for CFU counting. However, the lack of consistency and correlation between methodologies, which can lead to misleading interpretations when testing new drugs, drove us to use CFU counts as rendering the clearer, more reliable and reproducible results. Another technical aspect is the culture medium of choice. Although results obtained by bacterial growth in CDM or in sBHI were relatively similar, we consistently found differences for some strains when incubated with the highest drug doses tested, which may relate to the apparently biphasic mode of action suggested above (Fig. S5H to K). Lastly, the bacterial growth type of the culture may be another key aspect. Here, by following the same procedure, we monitored the effect of FabHi on bacteria in suspension and attached to a plastic surface. Differences were noticeable, as an inverse correlation was observed on the FabHi effect when assessing planktonic or biofilm-grown bacteria, as the lowest inhibitor concentration tested did not alter biofilm growth by those strains considered to be more susceptible, according to planktonic studies, compared to the more resistant ones. In this context, strain P657 showed slow growth in CDM, although it clearly grouped in the most resistant category despite low expression of the *fabH* gene, and

the effect of FabHi on P657 biofilm growth could not be easily explained, as it was consistently unrelated to the concentration of inhibitor used.

**FabHi effects are heterogeneous among NTHi clinical isolates.** Strain heterogeneity when incubated with FabHi evidenced the concurrence of several factors in its killing activity. According to the docking model, FabHi fills a large tunnel, at the base of which are located the active-site residues. This tunnel is the only access to the active center and, therefore, constitutes the binding cavity for the first (acyl-CoA) and second (malonyl-ACP) substrates and the feedback inhibitor (palmitoyl-ACP), all of which will compete with FabHi binding. Hence, the antimicrobial effect of FabHi depends on the available concentrations of all potential ligands and their respective affinities, which may vary among strains and growth conditions. Indeed, as shown above, several polymorphisms in FabH variants locate at regions relevant for activity and structural stability, and sequence changes at such positions may alter substrates and inhibitor binding as well as FabH catalytic efficiency. Additionally, strain variability associated with different CTs might modify, among others, the efflux pump system, with a subsequent impact on the amount of FabHi available to interact with FabH. Likewise, variations in expression level, conformation, or potency of other components of the FASII biosynthetic pathway, including peripheral enzymes involved in production of FabH substrates (78), may modify the catalytic balance of rates along the FASII pathway and, thus, alter the antimicrobial effect. Indeed, mutations might compensate for deficiencies in other enzymes of the FASII system, rescuing specific *H. influenzae* strains from FabHi antimicrobial effect or, alternatively, enhancing its outcome.

Nonetheless, variations observed among strains carrying the same FabH variant indicate that heterogeneity is multifactorial. Those strains are diverse in terms of genomic content, i.e., they do not cluster by CT (45), and such genomic variability may contain currently unidentified elements also contributing to the observed phenotypes.

On the other hand, inspection of FabHi:FabH$_{RdKW20}$ complex structure indicates that FabHi affinity could be increased by introducing a hydroxyl group *ortho* to the N-atom of the piperidine ring, which could lead to an extra hydrogen bond with the side chain of Asn246 without introducing steric clashes. While of interest for further analysis, exploring the effect of FabHi modifications exceeds the scope of this study.

Overall, we present here a high-quality metabolic model of *H. influenzae* and show how *i*NL638 can be used (i) as a powerful tool to better understand the metabolism of this pathogen, (ii) as a computational framework for large experimental data set contextualization, and (iii) for the discovery of condition-independent drug targets. We shed light on the mechanism of inhibition of FabH mediated by FabHi, and, beyond allelic variation, we suggest the strain-specific metabolic context is a key factor modulating drug performance. Therefore, the systematic metabolic evaluation of individual strains through computational frameworks will be of critical importance in the near future to design more precise and accurate treatments.

## MATERIALS AND METHODS

**Bacterial strains, media, growth conditions, and drugs.** RdKW20 is a genome-sequenced capsule-deficient laboratory strain (43). NTHi375 is a genome-sequenced otitis media clinical isolate (47–50). NTHi COPD clinical strains belong to a previously genome-sequenced collection (45). For growth in sBHI or CDM, NTHi strains grown on PolyViteX (PVX) agar for 16 h were inoculated (2 to 3 colonies) in 10 mL sBHI or CDM using 100-mL flasks and incubated for 11 h with shaking (100 rpm). In detail, 235 mL of CDM contains 191 mL of RPMI 1640 (Life Technologies, ref. 11879-020), 5.8 mL of 1 M Hepes (pH 7.2–7.5, Life Technologies, ref. 15630-080), 2 mL of 100 mM MEM sodium pyruvate (Life Technologies, ref. 113600-070), 10 mL of uracil (2 mg/mL), 20 mL of inosine (20 mg/mL), 2 mL of NAD (1 mg/mL), 4 mL of hemin (1 mg/mL), and 2.35 mL of 1 M glucose (final concentration, 10 mM). Cultures were then diluted to an optical density at 600 nm (OD$_{600}$) of 0.07 in a final volume of 25 mL sBHI or CDM using 250-mL flasks, grown with shaking (180 rpm), and OD$_{600}$ was recorded every hour for up to 8 h. When necessary, bacterial viability upon host cell infection conditions was tested. To do so, phosphate-buffered saline (PBS)-normalized bacterial suspensions (OD$_{600}$ of 1) were prepared by using NTHi grown on PVX agar; 100-$\mu$L aliquots were incubated in 1 mL Earle's balanced salts (EBSS; 24010-043; Gibco) in the absence/presence of FabH inhibitor (FabHi) for 2 h at 37℃, serially diluted, and plated on supplemented Haemophilus Test Medium (sHTM) agar for CFU/mL determination. When needed, media supplements refer to with 10 $\mu$g/mL hemin and 10 $\mu$g/mL nicotinamide adenine dinucleotide (NAD).

FabHi was used by freshly preparing a panel of stock solutions, customized to each type of assay (7 to 35 mg/mL) in dimethyl sulfoxide (DMSO), and diluted to the working concentrations specified in each assay in CDM, sBHI, or RPMI 1640 (R8758; Sigma-Aldrich). Antibiotics and FabHi working concentrations are specified for each assay (described below). Ampicillin and azithromycin dihydrate (Azm; Zytromax) were used to assess FabHi synergic effects (described below). For this purpose, 10-mg/mL stock solutions of each antibiotic were prepared in distilled $H_2O$ (d$H_2O$), filtered, and used at the indicated working concentrations.

**Determination of bacterial dry weight.** Two to 3 colonies of the RdKW20 strain grown on PVX agar for 16 h were inoculated in 10 mL CDM using 100-mL flasks and incubated for 11 h with shaking (100 rpm). Cultures were diluted to $OD_{600}$ of 0.07 in a final volume of 55, 40, or 25 mL CDM using 250-mL flasks, grown with shaking (180 rpm) for ~3, 4, or 6 h, respectively. Next, 45, 30, and 15 mL was collected at $OD_{600}$ of 0.3, 0.4 and 0.8, respectively. Bacterial suspensions were centrifuged at (20,000 × *g*, 15 min, room temperature), and pellets were instantly frozen in liquid $N_2$, directly transferred to a lyophilizer (LyoQuest, Telstar), lyophilized for 24 h at $0.020 \times 10^5$ Pa and $-80°C$, and weighed. Using this procedure, we established a lineal relation ($R^2$ = 0.99) between OD and DW: DW (g/liter) = (0.8627 × $OD_{600}$) + 0.0279 (see Table S1 in the supplemental material).

**Model construction and manual curation.** Since the available GEM of *H. influenzae* lacks the GPR relationship, it could not be used as the template for draft construction. Instead, we used the high-quality and well-curated GEM of *E. coli* K-12 (*i*JO1366 [79]) for this purpose. GEMSiRV-MrBac server was used to construct the protein homology matrix and the initial draft (80). Briefly, the list of orthologous genes between the RdKW20 and K-12 strains was obtained by reciprocal best hits using BLASTP (81). This list and the corresponding model of *E. coli* in SBML format were used as input files for the GEMSiRV-MrBac server to obtain the initial model draft of RdKW20. This initial draft included the reactions present in *i*JO1366 assigned to the orthologous genes identified in RdKW20. Subsequently, the draft was subject to a detailed curation by manually reviewing all the GPR associations. The assignment of each GPR was additionally validated by BLASTP and, when possible, by detailed scrutiny of available literature. Pathways for well-known metabolic features of strain RdKW20 and absent in the model template, such as strain-specific lipooligosaccharide (LOS) and peptidoglycan biosynthesis, were manually mass and charge balanced and added to the reconstruction. Furthermore, the reactions involving strain-specific cofactors were updated, such as those reactions using menaquinones or ubiquinones from the *E. coli* model, which were replaced by reactions accounting the cofactor usage of *H. influenzae*, i.e., demetylmenaquinone and demethylmenaquinol, respectively. Finally, the network gaps across the metabolic pathways were filled out by manual gap filling by adding new reactions based on the information stored in biological databases, scientific literature, and the functional annotation of strain RdKW20. For instance, metabolic gaps in fatty acid and molybdopterin biosynthesis pathways were manually completed during this step. Additionally, transport reactions were manually curated considering the connectivity of the cognate metabolites and experimental evidence. For instance, transport reactions for certain amino acids, fatty acids, uracil, pantothenate, biotin, dihydrofolate, protoporphyrin, pyridoxine, riboflavin, nucleotides, or NAD transporters were added manually based on available biochemical and physiological evidence. On the contrary, transport reactions for several compounds initially included in the automatic draft were removed due the lack of physiological evidence. The evaluation of *i*NL638 was done with MEMOTE v0.13.0 (https://memote.io/) (25).

**Biomass objective function (BOF) formulation.** For BOF construction, reported macromolecular composition of *H. influenzae* RdKW20 was used as a base for the construction of a highly complete biomass reaction (Data Set S1). Protein, DNA, RNA, LOS, and lipid contents were taken from available literature (82–84). Missing information, such as carbohydrates, inorganic ions, and soluble pool content, was taken from the *E. coli* model (79, 85, 86) (Data Set S1). The stoichiometric coefficients for amino acids, DNA, and RNA were further determined computationally using the *H. influenzae* available genomic information (Data Set S1), while the coefficients for murein and lipids were taken from experimental reports. In the absence of experimental information, growth-associated ATP maintenance reaction (GAM) was estimated by determining the energy required for macromolecular synthesis, as previously described (24), while the non-growth-associated ATP maintenance reaction (NGAM) was taken from the *E. coli* model (79).

**Constraint-based analysis.** The *i*NL638 model was analyzed with the COBRA Toolbox v2.0 (87) within the MATLAB environment (The MathWorks Inc.), and Gurobi and the GNU Linear Programming kit (http://www.gnu.org/software/glpk) were used for solving the linear programing problems. The constraint-based model consists of a 1,161 by 1,385 matrix containing all the stoichiometric coefficients in the model of 1,161 metabolites and 1,385 reactions (*S*). Flux balance analysis (FBA) was used to predict growth and flux distributions (88). FBA is based on solving a linear optimization problem by maximizing or minimizing a given objective function *Z*, subject to a set of constraints. The constraints *S·v* = 0 correspond to a situation of steady-state mass conservation where the change in concentration of the metabolites as a function of time is zero. The vector *v* represents the individual flux values for each reaction. These fluxes are further constrained by defining lower and upper limits for flux values. For reversible reactions, upper and lower bounds of $-1,000$ mmol·gDW$^{-1}$·h$^{-1}$ and 1,000 mmol·gDW$^{-1}$·h$^{-1}$ were used. A lower bound of 0 mmol·gDW$^{-1}$·h$^{-1}$ was used in the case of irreversible reactions. For simulating condition-specific growth conditions, lower bounds of the corresponding exchange reactions were modified accordingly.

**Model constraints.** Formulation of *in silico* CDM was based on the experimental composition of this medium. The uptake of pyruvate, glucose, NAD, inosine, and uracil was allowed as main nutrients in all cases (26, 27, 89). Furthermore, complete and minimal CDM formulations (cCDM and mCDM) were used. cCDM allowed the uptake of all amino acids, vitamins, and inorganic salts present in the RPMI formulation (https://www.thermofisher.com/es/es/home/technical-resources/media-formulation.116.html), whereas mCDM formulation only allowed uptake of inorganic salts and essential vitamins for growth (i.e., choline, pantothenate, and

folate). Source code to simulate the *i*NL638 with cCDM and mCDM media and the MEMOTE full report are available at https://github.com/SBGlab/Haemophilus_influenzae_GEM.

**Essentiality analysis.** Reaction essentiality analysis was performed using the *singleReactionDeletion* function implemented in the COBRA Toolbox. FBA (88) and minimization of metabolic adjustments (MOMA) (90) methods were used. A lethal deletion was defined as that yielding <10% of the original model's growth rate values. Simulations for reaction essentiality were performed using both cCDM and mCDM for *i*NL638 under aerobic conditions.

**Synthetic procedure for FabHi.** Please see Fig. 3A for analysis results of all compounds, in full accordance with depicted structures.

**(i) General methods.** $CH_3CN$ was dried by reflux over $CaH_2$. Anhydrous DMF and reagents were obtained from commercial sources and used without further purification. Analytical thin-layer chromatography (TLC) was performed on silica gel 60 (F254; Merck). Compounds were purified by flash column chromatography with silica gel 60 (230 to 400 mesh) (Merck). The purity of the compounds was analyzed using a high-performance liquid chromatography mass spectrometer (HPLC-MS) and was performed on an HPLC Waters 2695 instrument connected to a Waters Micromass ZQ 2000 spectrometer and a photodiode array detector. The column used was a Sunfire $C_{18}$ (4.6 mm by 50 mm, 3.5 mm), and the flow rate was 1 mL min$^{-1}$. All retention times are quoted in minutes. Nuclear magnetic resonance (NMR) spectra were recorded with Varian XL-400 spectrometer operating at 400 MHz for $^1$H-NMR and at 100 MHz for $^{13}$C with $Me_4Si$ as an internal standard. Mass spectra were measured on a quadrupole mass spectrometer equipped with an electrospray source (LC/MC HP1100; Hewlett-Packard). Microanalyses were obtained on a Heraeus CHN-O-RAPID instrument. The purity of novel compounds was also determined to be >95% by elemental analysis.

**(ii) Ethyl 1-(5-bromopyridin-2-yl)piperidine-4-acetate (1).** A solution of ethyl piperidine-4-acetate (1.5 g, 8.76 mmol), 5-bromo-2-chloropyridine (2 g, 10.39 mmol), and triethyl amine (1.47 mL, 1.2 eq) in dry acetonitrile was heated under microwave irradiation to 125°C for 2 h. The crude was diluted with ethyl acetate (EtAcO) (10 mL) and washed with 1N HCl. The combined organics were dried over $MgSO_4$, filtered, and evaporated to dryness. The final residue was purified by flash column chromatography (hexane-ethyl acetate, 1:3 to 0:1) to give 1 (1.15 g, 40%) as a colorless oily solid. $^1$H-NMR (400 MHz, DMSO-$d_6$) $\delta$ 8.19 (d, J = 2.7 Hz, 1H), 7.76 (dd, J = 9.2, 2.6 Hz, 1H), 6.95 (d, J = 9.2 Hz, 1H), 4.00 (m, 4H), 2.92 (td, J = 12.9, 2.7 Hz, 2H), 2.35 (dd, J = 8.8, 7.0 Hz, 3H), 1.84 (d, J = 13.3 Hz, 2H), 1.76 (d, J = 12.8 Hz, 1H), 1.52 − 1.41 (m, 1H), 1.23 (t, J = 7.0 Hz, 3H). HPLC 9.02 min (98%) ($H_2O/CH_3CN$ from 15/85 to 95/5 in 10-min flow-rate of 1 mL/min). MS (ES+) *m/z* 327.0 (M)$^+$, 329.0 (M + 2)$^+$. Anal. for $C_{14}H_{19}BrN_2O_2$ (C, H, N): C, 51.39; H, 5.85; N, 8.56. Found: C, 51.22; H, 5.91; N, 8.63.

**(iii) Ethyl 1-(5-(2-fluoro-5-(hydroxymethyl)phenyl)pyridin-2-yl)piperidine-4-acetate (2).** A suspension of 1 (1.15 g, 3.68 mmol), 2-fluoro-5-(hydroxymethyl)phenylboronic acid (0.89 g, 5.26 mmol), [1,1'-bis(diphenylphosphino)ferrocene]palladium(II) dichloride (172 mg, 0.21 mmol), $K_2CO_3$ (0.89 g, 7.02 mmol) in anhydrous DMF (12 mL) was stirred at 75°C for 16 h. The reaction was cooled to room temperature, and the residue was filtered through Celite and evaporated to dryness. The final residue was purified by flash column chromatography (hexane-ethyl acetate, 5:1 to 1:2) to give 2 (1.15 g, 87%) as a white solid. $^1$H-NMR (400 MHz, DMSO-$d_6$) $\delta$ 8.40 (s, 1H), 7.71 (d, J = 8.9 Hz, 1H), 7.39 (dd, J = 1.9, 7.5 Hz, 1H), 7.25 (m, 1H), 7.13 (dd, J = 8.5, 10.5 Hz, 1H), 6.73 (d, J = 9.0 Hz, 1H), 4.75 (s, 2H), 4.28 (dt, J = 13.3, 3.7 Hz, 2H), 4.17 (m, 2H), 3.12 (t, J = 11.1 Hz, 2H), 2.54 (m, 1H), 2.00 (m, 2H), 1.83 (m, 3H), 1.27 (t, J = 7.2 Hz, 3H). HPLC 5.03 min (99%) ($H_2O/CH_3CN$ from 15/85 to 95/5 in 10-min flowrate of 1 mL/min). MS (ES+) *m/z* 373.2 (M + 1)$^+$. Anal. for $C_{21}H_{25}FN_2O_3$ (C, H, N): C, 67.72; H, 6.77; N, 7.52. Found: C, 67.51; H, 6.80; N, 7.65.

**(iv) FabHi.** A solution of 2 (1.15 g, 3.21 mmol) in THF-methanol (MeOH) (1:1, 36 mL) and 1 M NaOH (1.5 mL) was refluxed for 4 h. The mixture was evaporated to dryness, and the residue was dissolved in a mixture of EtAcO-$H_2O$ (1:1, 10 mL). The aqueous phase was acidified to pH 5, and the organic phase was separated. The aqueous phase extracted with EtAcO, and the organic phase combined was dried over $MgSO_4$, filtered, and evaporated to dryness to obtain 3 (0.85 g, 80%). $^1$H-NMR (400 MHz, DMSO-$D_6$) $\delta$ 8.24 (t, J = 2.1 Hz, 1H), 7.65 (dt, J = 8.9, 2.0 Hz, 1H), 7.38 (dd, J = 8.0, 2.2 Hz, 1H), 7.22 (m, 2H), 6.87 (d, J = 8.9 Hz, 1H), 4.47 (s, 2H), 4.30 (dt, J = 13.1, 3.3 Hz, 2H), 2.80 (td, J = 12.7, 2.6 Hz, 2H), 2.14 (d, J = 7.0 Hz, 2H), 1.90 (m, 1H), 1.70 (m, 2H), 1.12 (m, 2H). $^{13}$C-NMR (100 MHz, DMSO-D6) $\delta$ 174.1, 158.6 (d, $J_{C-F}$ = 243.3 Hz), 158.5, 147.7 (d, J = 4.2 Hz), 139.7 (d, $J_{C-F}$ = 3.3 Hz), 138. 1 (d, $J_{C-F}$ = 3.2 Hz), 128.3 (d, $J_{C-F}$ = 3.6 Hz), 127.2 (d, $J_{C-F}$ = 8.2 Hz), 125.6 (d, $J_{C-F}$ = 13.8 Hz), 119.8, 116.2 (d, $J_{C-F}$ = 22.9 Hz), 107.1, 62.7, 60.3, 45.2, 41.2, 33.3, 31.5. HPLC 1.35 min (96%) ($H_2O/CH_3CN$ from 15/85 to 95/5 in 10-min flowrate of 1 mL/min). MS (ES+) *m/z* 345.4 (M + 1)$^+$. Anal. for $C_{19}H_{21}FN_2O_3$ (C, H, N): C, 66.27; H, 6.15; N, 8.13. Found: C, 66.31; H, 6.05; N, 8.19.

**FabHi docking studies.** Docking studies were performed with the AutoDock 4.2 (91) and AutoDock Vina (92) programs using the atomic coordinates of NTHi RdKW20 protein (FabH$_{RdKW20}$; Protein Data Bank entry 3IL3) as receptor and FabHi as ligand. In detail, the protein was considered rigid and the ligand flexible. FabHi structure was carefully built using the PyMOL visualizer (PyMOL Molecular Graphics System, version 1.5.0.4; Schrödinger, LLC) from the atomic coordinates of ligand cocrystallized with *E. coli* FabH in PDB entry 5BNR (compound 23 in reference 38), which lacks only the methylene unit of the FabHi carboxylic moiety, followed by molecule optimization and energy minimization using the PRODRG2 server (93). Ligand and protein files were edited and prepared with the AutoDockTools 1.5.6 program (91) after substituting dicysteine derivative at position 112 in FabH$_{RdKW20}$ structure by cysteine using PyMOL. Polar hydrogens and Kollman charges were added to FabH$_{RdKW20}$ structure. Gasteiger charges were computed for the ligand, whose active torsion angles were allowed to rotate during docking. For docking simulations, 200 Lamarckian genetic algorithm simulation runs were performed with 25 million energy evaluations per run. The best docking solutions were selected by clustering within the default 2.0-Å root mean square deviation value and ranking the largest cluster solutions by the free

energy AutoDock scoring function. The binding mode with lowest docking energy and best optimized configuration of catalytic triads was processed by adding hydrogens and minimizing *in vacuo* by 2,500 cycles of conjugate gradient followed by 500 cycles of steepest descent using the Sander program (94, 95). The minimized solution was selected as the most probable model of the $FabH_{RdKW20}$-FabHi complex.

**Determination of FabHi antimicrobial effects. (i) Dose-dependent FabHi antibacterial effect under bacterial planktonic or biofilm growth.** A 25-mg/mL FabHi stock solution in DMSO was used to prepare 40-, 200-, 300-, 400-, 500-, 600-, 800-, 1,000-, 1,100-, 1,200-, 1,600-, 2,000-, and 2,400-$\mu$g/mL working solutions in CDM or in sBHI. Fifty-microliter aliquots of sBHI or CDM were transferred to individual wells. Next, 50 $\mu$L of each FabHi working solution was added and mixed and 50 $\mu$L was discarded. A suspension of PVX agar with freshly grown bacteria was generated in sBHI or CDM, adjusted to 0.5 MacFarland ($OD_{600}$ of 0.063) and diluted 1:100. Next, 50-$\mu$L bacterial aliquots were transferred to each well. A vehicle solution control consisting of a volume of DMSO equivalent to that of the highest FabHi concentration tested was performed in parallel (indicated as 0 [FabHi] $\mu$g/mL). Blank controls (sBHI or CDM added) were used. Plates were incubated for 24 h at 37°C, 5% $CO_2$, without shaking. Culture samples were serially diluted and plated on sHTM agar. Data are shown as log CFU/mL; experiments were performed in duplicate at least three times ($n \geq 6$). Alternatively, FabHi antimicrobial effect was assessed on *H. influenzae* biofilm grown in sBHI. For this purpose, assays were set as indicated above, and bacterial growth was determined by measuring $OD_{600}$ on a SpectraMax 340 microplate reader. The liquid portion in each well (containing planktonic bacteria) was then discarded, and plates were washed 3 times through gentle submersion in distilled water and allowed to air dry. Next, 150 $\mu$L/well 0.5% crystal violet dissolved in $dH_2O$ was added, and plates were incubated for 20 min with gentle shaking at room temperature, followed by plate washing as previously described. Finally, 150 $\mu$L/well 95% ethanol (Merck) was added, plates were incubated for 20 min with gentle shaking at room temperature, and the $OD_{570}$ was determined as a measure of biofilm growth. The $OD_{570}/OD_{600}$ ratio for each strain and independent assay was calculated to normalize biofilm biomass to overall growth; experiments were performed in triplicate at least three times ($n \geq 9$).

**(ii) Antimicrobial synergic effects.** A 7-mg/mL FabHi stock solution was used to prepare 300-$\mu$g/mL working solutions in sBHI; 10-mg/mL Amp or Azm stock solutions were used to prepare 4-$\mu$g/mL and 2-$\mu$g/mL working solutions in sBHI, respectively. After generating each FabHi-Amp or FabHi-Azm concentration matrix, each well of a 96-well microtiter plate contained 80 $\mu$L final volume. A suspension of PVX agar and freshly grown bacteria was generated in sBHI, adjusted to $OD_{600}$ of 0.16, and diluted 1:100, and 20-$\mu$L aliquots were transferred to the plate wells. Plates were incubated for 24 h at 37°C, 5% $CO_2$, without agitation. The fractional inhibitory concentration (FIC) index of each FabHi-Amp or FabHi-Azm combination was calculated to determine the existence of synergy ($\Sigma FIC \leq 0.5$), additive ($0.5 > \Sigma FIC \leq 1$), indifferent ($1 > \Sigma FIC < 4$), or antagonic ($\Sigma FIC \geq 4$) effects. Three independent assays were performed ($n = 3$).

**(iii) Serial passage experiments with FabHi.** A 14-mg/mL FabHi stock solution was used to prepare 550-, 1,100-, and 2,200-$\mu$g/mL working solutions in sBHI. A vehicle solution control (indicated as V) consisting of a volume of DMSO equivalent to that of the highest FabHi concentration tested was performed in parallel. Blank controls (sBHI) were used; 50 $\mu$L sBHI was transferred to individual wells and mixed with 50 $\mu$L of each FabHi working or vehicle control solution, and 50 $\mu$L was discarded. A suspension of PVX agar and freshly grown bacteria was generated in sBHI, adjusted to 0.5 MacFarland, and diluted 1:100. Next, 50-$\mu$L bacterial aliquots were added to individual wells. Plates were incubated for 24 h at 37°C, 5% $CO_2$, without shaking. Cultures were then passaged (50 $\mu$L in 50 $\mu$L fresh sBHI with FabHi or vehicle control) every day for 12 days. At each time point throughout the cycling, each well's absorbance ($OD_{600}$ at 24 h minus $OD_{600}$ at 0 h) was measured. Nine replicates per condition were made ($n = 9$).

**Bacteria gene expression analyses.** Two to three colonies of bacterial strains grown on PVX agar were inoculated in 10 mL CDM or sBHI, grown for 11 h, diluted into 25 mL fresh CDM or sBHI to $OD_{600}$ of 0.07, and grown to half of each strain's maximal $OD_{600}$ for ~5 h. Bacterial total RNA was isolated using TRIzol reagent (Invitrogen) and total RNA quality evaluated using RNA 6000 Nano LabChips (Agilent 2100 Bioanalyzer, Santa Clara, CA). Reverse transcription (RT) was performed using 1 $\mu$g RNA by PrimerScript RT reagent kit (TaKaRa). To amplify *fabH*, 1:10 diluted cDNA was used as the template (including *gyrA* endogenous control). In all cases, 20-$\mu$L reaction mixtures containing 1× SYBR Premix *Ex Taq* II (Tli RNaseH Plus) (TaKaRa) and the adequate primer mix were used. Fluorescence was analyzed with an AriaMx real-time PCR system (Agilent Technologies). The comparative threshold cycle ($C_T$) method was used to obtain relative quantities of mRNA that were normalized using *gyrA* as an endogenous control. The *fabH* gene was amplified with primers *fabH*-F and *fabH*-R in all cases except for the P851 and P642 strains, where *fabH*-P851-P642-F and *fabH*-P851-P642-R primers were used (Table S2). Samples were grown in triplicate and processed with technical triplicates ($n = 3$).

**Infection of cultured cells.** A549 cells were maintained as described previously (48) and seeded on 24-well tissue culture plates to $1.5 \times 10^5$ cells/well for 32 h. To assess FabHi cytotoxicity, a 35-mg/mL FabHi stock solution was diluted in serum-starved RPMI 1640 (Sigma-Aldrich) to 125, 275, and 500 $\mu$g/mL (1 mL/well), cells were incubated for 16 h at 37°C, 5% $CO_2$, and cytotoxicity was determined by measuring the release of lactate dehydrogenase (CytoTox 96; Promega) and microscopy. To assess the effect of FabHi during epithelial infection by NTHi, cells were maintained and seeded as indicated above and treated with FabHi in serum-starved RPMI 1640 for 16 h before infection. For infection, PBS-normalized bacterial suspensions ($OD_{600}$ of 1) were prepared by using NTHi grown on PVX agar. Adhesion and invasion assays were performed as previously described (47, 48). Controls (indicated as V in the figures) were performed by using a DMSO volume corresponding to that of the highest FabHi concentration tested. After infection, wells were washed and cells lysed as previously described (48). Lysates were serially

diluted in PBS and plated on sHTM agar for bacterial counts. Results are expressed as number of CFU/well. Experiments were performed in triplicate on at least three independent occasions ($n = 9$).

**NTHi adult zebrafish infection.** Animal experiments conducted at the Ikan Biotech (https://www.ikanbiotech.com) animal housing facility were performed as previously described (58), according to the approval of the Universidad de Navarra (UNAV) Ethics Committee for Animal Experimentation (protocol 107-19). To determine the maximum tolerated dose of FabHi in adult zebrafish, a toxicity assay was first performed in zebrafish embryos by following the guideline OECD TG236 (96) (Fig. S6B). Afterwards, 6-month-old zebrafish 0.30 $\pm$ 0.08 g in weight were randomly divided into 2 infected and 2 uninfected groups ($n = 10$/group). Infected groups were intraperitoneally injected with 10 $\mu$L of an exponentially grown (OD$_{600}$ of 0.3) *H. influenzae* suspension containing $7 \times 10^{10}$ CFU/mL ($7 \times 10^8$ CFU/zebrafish) prepared in PBS. At 1 hpi, one infected and one uninfected group were intraperitoneally administered FabHi at a dose of 4 $\mu$g/zebrafish in 10 $\mu$L PBS by using a 35 mg/mL FabHi stock solution; the other two groups were administered PBS or vehicle solution. The survival rate for each group was monitored three times per day for 4 days after infection. Experiments were performed on three independent occasions.

**Statistical analyses.** In all cases, a *P* value of <0.05 was considered statistically significant. Statistical analyses were performed using Prism software, version 7, for Mac (GraphPad Software, San Diego, CA, USA), and are detailed in each figure legend.

## SUPPLEMENTAL MATERIAL

Supplemental material is available online only.

**DATA SET S1**, XLSX file, 0.1 MB.
**DATA SET S2**, XLSX file, 0.1 MB.
**FIG S1**, TIF file, 2.6 MB.
**FIG S2**, TIF file, 1.6 MB.
**FIG S3**, TIF file, 2.8 MB.
**FIG S4**, TIF file, 2.8 MB.
**FIG S5**, TIF file, 2.7 MB.
**FIG S6**, TIF file, 0.5 MB.
**TABLE S1**, DOCX file, 0.03 MB.
**TABLE S2**, DOCX file, 0.1 MB.

## ACKNOWLEDGMENTS

We are grateful to Ariadna Fernández-Calvet and David C. McKinney for technical support. N.L.-L. is funded by a Ph.D. studentship from Regional Navarra Govern, Spain, reference 0011-1408-2017-000000. This work has been funded by grants from MICIU/AEI RTI2018-096369-B-I00, 875/2019 from SEPAR, PC150-151-152 from Gobierno de Navarra to J.G., by PID2019-108458RB-I00 MCIU/AEI /10.13039/501100011033 to J.N., and by RTI2018-099985-B-I00 to M.M. CIBER is an initiative from Instituto de Salud Carlos III (ISCIII), Madrid, Spain.

Conceptualization, J.G., J.N., M.M., M.J.C.; experimental work, N.L.-L., D.S.L., J.N., S.D.C., R.D.-M., M.I.-B.; data analysis, N.L.-L., D.S.L., J.N., R.D.-M., M.I.-B., M.M., J.G.; manuscript writing, N.L.-L, M.J.C., M.M., J.N., J.G.; manuscript review and editing, all authors; supervision, J.G.; funding acquisition, J.G., J.N., M.M.

We have no declarations of interest.

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
