## [Reviewer comments · mSystems]

Interrogation of essentiality in the reconstructed *Haemophilus influenzae* metabolic network identifies lipid metabolism antimicrobial targets: preclinical evaluation of a FabH β -ketoacyl-ACP synthase inhibitor.

Nahikari López-López, David San León, Sonia de Castro, Roberto Díez-Martínez, Manuel Iglesias-Bexiga, María-José Camarasa, Margarita Menéndez, Juan Nogales, and Junkal Garmendia

Corresponding Author(s): Junkal Garmendia, CSIC

Review Timeline:

Submission Date:	December 9, 2021
Editorial Decision:	January 6, 2022
Revision Received:	January 28, 2022
Accepted:	February 10, 2022

Editor: Youjun Feng

Reviewer(s): Disclosure of reviewer identity is with reference to reviewer comments included in decision letter(s). The following individuals involved in review of your submission have agreed to reveal their identity: Hai-Hong Wang (Reviewer #1); Lei Zhu (Reviewer #2)

Transaction Report:

DOI: <https://doi.org/10.1128/msystems.01459-21>

January 6, 2022

Dr. Junkal Garmendia
CSIC
Instituto de Agrobiotecnología
Avenida Pamplona 123
Mutilva, Navarra 31192
Spain

Re: mSystems01459-21 (Interrogation of essentiality in the reconstructed *Haemophilus influenzae* metabolic network identifies lipid metabolism antimicrobial targets: preclinical evaluation of a FabH β -ketoacyl-ACP synthase inhibitor.)

Dear Dr. Junkal Garmendia:

Thank you for submitting your manuscript to mSystems. We have completed our review and I am pleased to inform you that, in principle, we expect to accept it for publication in mSystems. However, acceptance will not be final until you have adequately addressed the reviewer comments.

Preparing Revision Guidelines

Sincerely,

Youjun Feng

Editor, mSystems

Journals Department
Reviewer comments:

Reviewer #1 (Comments for the Author):

This article constructed a high-quality genome-scale metabolic model with human pathogen *Haemophilus influenzae*, and figured out lots of essential genes involved in lipid metabolism. Authors then used computational and experiment work to evaluate the reported chemical compound FabHi, which inhibited the 3-ketoacyl ACP synthase III FabH in fatty acid synthesis. The results showed FabHi reduced *H. influenzae* viability in a dose-dependent manner, but different strains shown different inhibitory manner with no relationship with fabH expression level. FabHi showed no cytotoxicity to animal cells and protective effect on zebrafish systemic infection with NTHi.

1. In methods part "Determination of bacterial dry weight", the formula: $DW (g/L) = OD600 \times 0.9097$ is difficult to understand, and is not consistent with the results in Table S1. Moreover, it is not clear how the formula used in the whole text.
2. Line 349 "Fourteen different fabH allelic variants were found (alleles 1 to 14, i.e. A1-A14) ..." How did the author classify all the variants into 14 groups? What are the differences between different groups?
3. Line 370, A13 allelic variant was in different categories based on the inhibition grown on different media. Why? the authors can give some explanations?
4. Line 420-422, "sequence changes in any of these positions may have a significant impact on the functioning of the FabH enzyme, including affinities for" The authors confirmed that fabH gene is essential for *Haemophilus influenzae* with bioinformatic or experimental methods. Are there some evidences to prove the site-mutations affect the FabH function or bacterial growth?
5. Line 455-458, for Figure 6A, the authors should show some published references for the method to evaluate the induced bacterial resistance, to prove that the results were reliable. Moreover, in Figure 6A, what did the "CON" and "Vehicle" mean?

Reviewer #2 (Comments for the Author):

It is not surprising that metabolic modeling was used for antimicrobial target discovery, and it is a long time since fatty acid biosynthesis enzymes was reported as an antimicrobial target. However, manuscript entitled "Interrogation of essentiality in the reconstructed *Haemophilus influenzae* metabolic network identifies lipid metabolism antimicrobial targets: preclinical evaluation of a FabH -ketoacyl-ACP synthase inhibitor" brings a complete and clear story about the model (iNL638) and the drug (FabHi). The following are the questions and some mistakes in this manuscript:

1. Natural chemically synthesized inhibitors for FabH have been reported. But there is no mention that why this search focused your attention on 1-(5-(2-Fluoro-5-(hydroxymethyl)phenyl)pyridin-2-yl)piperidine-4-acetic acid. So what is the special advantage of this drug?
2. Since most of lipid metabolic genes in fig 4 was red and plsC gene was marked as an essential gene in Wong et al., 2013, PNAS, and Mobegi et al., 2014, BMC Genomics, How to understand that plsC gene was not in table1 or essential gene list in iNL638 model?
3. fabH genes are conserved in variety species of bacteria. However, it is nonessential gene in some strain (like *Staphylococcus aureus*). And FabHi effects are heterogeneous among NTHi clinical isolates. So do you think bacteria is readily to develop a resistance to this antibiotic?
4. Line 176, 'a toxicity assay was performed in zebrafish embryos'. For the toxicity assay in zebrafish, the guideline of FET test was followed. However, six-month-old zebrafish was used instead of embryos.
5. Line 288 and 289, PlsB and PlsC, not 'Psi'.
6. Line 989 and 990, 'FabHi' is not the same as the 'FabHi' in figure legend in picture file in. TIF format.
7. Line 571, the references (Campbell and Cronan 2001) was not in the same format with others.
8. For fig6 F, the black inverted triangles were not shown in the fig.

Dear Sir/Madam,

We appreciate very much the revision and constructive comments on our manuscript mSystems01459-21, entitled "Interrogation of essentiality in the reconstructed *Haemophilus influenzae* metabolic network identifies lipid metabolism antimicrobial targets: preclinical evaluation of a FabH β -ketoacyl-ACP synthase inhibitor".

Following all indications, we have modified the manuscript, thus answering to the best of our capacity all the issues raised by both reviewers.

Please find next a point-by-point answer to the comments, and find attached the revised Main text, Supplementary Material and Figures.

Moreover, we have reorganized the supplementary material following indications, and figure and table numbering has been modified accordingly.

Reviewer #1:

Comments are very much appreciated. We next reply to the issues raised:

1. Methods "Determination of bacterial dry weight", the formula: $DW (g/L) = OD_{600} \times 0.9097$ is difficult to understand, and is not consistent with the results in Table S1. It is not clear how the formula used in the whole text.

We thank the reviewer for his/her comment. After analysing our data, we acknowledge some discrepancies between the formula and data presented in **Table S1**. This is due an error during data visualization. However, please take into account that this discrepancy does not really significantly change the general formula correlating DW and OD in *H. influenzae*. The corrected formula is $DW(g/L) = (0.8627 \times OD_{600}) + 0.0279$ (see below).

Overall, we have:

- i) included a new column in **Table S1**, showing DW mg/ml as a function of OD to facilitate the understanding of the formula relating OD and DW in *H. influenzae*;
- ii) updated the formula in the methods section: "Using this procedure, we established a lineal relation ($R^2=0,99$) between OD and DW as it follows: $DW (g/L) = (OD_{600} \times 0.8627) + 0.0279$ (**Table S1**)".

Modified **Table S1**.

OD_{600}	Collected volume (mL)	Dry pellet weight (mg)	mg/mL
0.3	45	12.05	0.27
0.4	30	11.9	0.40
0.8	15	10.7	0.71

Formula estimation.

On the other hand, the reviewer is correct when saying that the formula was not finally used in the actual manuscript. However, we consider that this is a critical piece of information for future uses/users of this newly presented model, and that it should be provided in this manuscript.

2. Line 349 "Fourteen different fabH allelic variants were found (alleles 1 to 14, i.e. A1-A14)...": How did the author classify all the variants into 14 groups? What are the differences between different groups?

As indicated in the Results section "FabHi has an antimicrobial effect on *H. influenzae* growth", we examined sequence variability of the *fabH* gene across a previously genome sequenced set of clinical strains (please see our previous work, mBio. 2018 Sep 25;9(5):e01176-18. doi: 10.1128/mBio.01176-18.).

We have extracted the best hits to *fabH* gene from this complete genome set. Next, genes were translated and resulting proteins were aligned. In this way, we generated a multiple alignment allowing us to compare FabH protein sequence in the complete strain set. The RdKW20 and NTHi375 reference strains were also included in this comparison.

This multiple comparison rendered fourteen different FabH sequence variants. We numbered them alleles (A) 1 to 14, i.e. A1-A14; numbering is arbitrary. **Figure S3** (numbering in revised version) shows a multiple alignment including FabH sequence for each of the 14 identified variant types, corresponding to that from one representative strain per FabH variant type.

We also analysed the distribution of this FabH variation across the above-cited collection of NTHi clinical isolates. Summary of such distribution is shown in **Table 3** (numbering in revised version). For completion, in our previous work we comparatively analysed the above-cited strain collection genomes and established clonal types (CT) based on their closeness. The distribution of FabH variation across those previously established CTs was analysed, with different frequency and distribution among CTs. This piece of information does not have a significant impact on strain selection and results obtained.

3. Line 370, A13 allelic variant was in different categories based on the inhibition grown on different media. Why? the authors can give some explanations?

This is true and it applies to four strains, A2-P621, A7-P665, A9-P645 and A13-P590. Differences between using sBHI and CDM media were mostly observed when testing high doses of the inhibitor (see **Figure S5H-K** data when using inhibitor concentrations higher than 300 µg/mL). These results were clearly reproducible and led us to allocate strains in different groups depending on assay media conditions. We acknowledge that do not know the reason for these differences, and cannot properly speculate by comparing media composition as it is not defined for BHI.

Although not fully conclusive, we considered important to show these data to call the reader's attention to experimental details, which are very important in this type of non-standardised assays.

4. Line 420-422, "sequence changes in any of these positions may have a significant impact on the functioning of the FabH enzyme, including affinities for...." The authors confirmed that fabH gene is essential for Haemophilus influenzae with bioinformatic or experimental methods. Are there some evidences to prove that site-mutations affect FabH function or bacterial growth?

We appreciate this comment.

We should explain that we have invested a fair amount of time and resources trying to establish possible associations between sequence polymorphisms and drug effects. Among others, we tried *fabH* gene variant replacement, aiming to exchange the *fabH*

gene from a more sensitive strain by the *fabH* variant from a more resistant strain. However, we did not manage to generate those strains. In any case, when comparing strains presenting the same FabH variant, we could not observe any clear association between variant and phenotype, and decided to stop those approaches. Our observations suggest that the FabH variant relates to the observed inhibitory phenotype but may not be the only factor, thus explaining why in some cases, strains with the same variant do not show identical phenotypes. At this point, we cannot provide information about possible additional factors.

Regarding a possible effect of FabH variants on bacterial growth, it seems to be unlikely or at least unclear. For example, strain P657 (highly resistant) does not grow particularly well in CDM but this is not the case in sBHI (see **Figure S5A**), although the resistant phenotype is observed independently of the assay medium.

We should also mention that it has been previously reported that single point mutations at positions predicted to interact with the inhibitor (Ala215; Ala245) increased the resistance of *H. influenzae* to a close structural analogue of FabHi (ACS Infect Dis. 2016 Jul 8;2(7):456-64. doi: 10.1021/acsinfecdis.6b00053.). As shown in **Figure S3**, these positions are conserved in the variants included in this study, and our predictions do not suggest Ala215 or Ala245 to be close to positions displaying polymorphisms.

In sum, we cannot provide clear explanations proving that site-mutations affect FabH function of bacterial growth.

5. Line 455-458, Figure 6A, the authors should show some published references for the method to evaluate the induced bacterial resistance, to prove that the results were reliable. Moreover, in Figure 6A, what did the "CON" and "Vehicle" mean?

We assessed the ability of NTHi to become resistant to FabHi through serial passage in broth containing the drug of interest by following a standard method (J Antimicrob Chemother. 2017 Jan;72(1):115-127. doi: 10.1093/jac/dkw381.). Moreover, we adapted this method to *H. influenzae* in a previous work (Biomolecules. 2019 Dec 17;9(12):891. doi: 10.3390/biom9120891.). We have updated the main text including these references.

Regarding nomenclature in **Figure 6**, this comment is very much appreciated. When revising this figure and legend, we see that it was unclear. CON refers to the assay positive control, i.e. bacteria grown in sBHI; vehicle refers to bacteria grown in sBHI in the presence of DMSO (vehicle solution for FabHi), by using a DMSO volume corresponding to that employed with the highest FabHi concentration tested in this assay, 550 µg/mL. For clarity, we have now modified figure and figure legend.

Reviewer #2:

Comments are very much appreciated. We next reply to the issues raised:

1. Natural chemically synthesized inhibitors for FabH have been reported. But there is no mention that why this search focused your attention on 1-(5-(2-Fluoro-5-(hydroxymethyl)phenyl)pyridin-2-yl)piperidine-4-acetic acid. So what is the special advantage of this drug?

Reasons to select this molecule are indicated in the main text, please see: This compound exhibited a potent inhibition of *H. influenzae* FabH activity ($IC_{50}=0.82 \mu M$), high solubility, acceptable human plasma protein binding, easy chemical accessibility (ACS Infect Dis. 2016 Jul 8;2(7):456-64. doi: 10.1021/acsinfecdis.6b00053), feasibility of visualizing its mode of binding by molecular docking and therefore improvement of favorable contacts through rational design.

2. Since most of lipid metabolic genes in Fig. 4 was red and plsC gene was marked as an essential gene in Wong et al., 2013, PNAS, and Mobegi et al., 2014, BMC Genomics, How to understand that plsC gene was not in Table1 or essential gene list in iNL638 model?

We appreciate this comment, that may contain a misunderstanding due to nomenclature issues. The *plsC* gene ID is HI0734. This gene was shown to be essential in our iNL638 model (see revised **Dataset S2**), and also in studies by Gawronsky (PNAS2009) and by Wong (PNAS2013), but not in the study by Mobegi (BMC Genomics). For this reason, as not being commonly essential in all the available studies, it is not included in **Table 2** and it is not labelled in red in **Figure S2**.

Please take into account that this apparent discrepancy is due alternative nomenclature for genes used in the different studies. To correct this and other similar discrepancies that we have detected, we have systematically updated **Datasheet S2** with the more recent nomenclature for the *H. influenzae* genes.

3. fabH genes are conserved in variety species of bacteria. However, it is nonessential gene in some strains (Staphylococcus aureus). And FabHi effects are heterogeneous among NTHi clinical isolates. So do you think bacteria is readily to develop a resistance to this antibiotic?

Please note that FASII inhibition may have different effects when assessing drug suitability against Gram positive or Gram negative bacteria. The essentiality of the FASII genes in Gram-negative bacteria arises from the requirement for β -hydroxy-fatty acids to assemble the lipid A core structure of LPS. Fatty acid supplementation cannot support lipid A synthesis because there is no mechanism to transfer acyl chains from CoA to the acyl carrier protein (ACP) of FASII so that the hydroxyl group can be introduced. Supplementation with hydroxy-fatty acids also is ineffective because the acyltransferases of lipid A biosynthesis use only ACP thioester substrates. However, this is not the case for Gram positive bacteria as they do not contain lipid A and LPS. Therefore, the feasibility of targeting FASII should not be identically considered for Gram positive and Gram negative bacteria. This also relates to differences in gene essentiality among bacterial species.

As indicated by the reviewer, FabHi effects were shown to be heterogeneous among strains, and we could not provide a mechanistic explanation for the observed heterogeneity. However, this may not necessarily mean that bacteria are ready to develop resistance. In fact, data shown in **Figure 6A** do not suggest resistance development in a serial passage assay. This is an in vitro assay and we acknowledge its limitations, but results were completely clear in this case.

4. Line 176, 'a toxicity assay was performed in zebrafish embryos'. For the toxicity assay in zebrafish, the guideline of FET test was followed. However, six-month-old zebrafish was used instead of embryos.

The efficacy trials were carried out in adult zebrafish. However, to determine toxicity, toxicity was tested in zebrafish embryos as required by the guidelines. For completion, a Figure is now added in the Supplementary Material showing the toxicity studies carried out, please see **Figure S6B**. With this assay, we were able to determine which is the maximum tolerated dose and therefore the one that should not be exceeded in adults.

5. Line 288 and 289, PlsB and PlsC, not 'Psi'.

Modified.

6. Line 989 and 990, 'FabHi' is not the same as the 'FabHi' in figure legend in picture file in. TIF format.

This has been checked and homogeneized.

7. Line 571, the reference (Campbell and Cronan 2001) was not in the same format with others.

Modified.

8. For Fig. 6 F, the black inverted triangles were not shown in the figure.

Please note that all four symbols are shown in the figure. White circles, triangles and inverted triangles are overlapping, as survival was 100% in these three cases.

February 10, 2022

Dr. Junkal Garmendia
CSIC
Instituto de Agrobiotecnología
Avenida Pamplona 123
Mutilva, Navarra 31192
Spain

Re: mSystems01459-21R1 (Interrogation of essentiality in the reconstructed *Haemophilus influenzae* metabolic network identifies lipid metabolism antimicrobial targets: preclinical evaluation of a FabH β -ketoacyl-ACP synthase inhibitor.)

Dear Dr. Junkal Garmendia:

Your manuscript has been accepted, and I am forwarding it to the ASM Journals Department for publication. For your reference, ASM Journals' address is given below. Before it can be scheduled for publication, your manuscript will be checked by the mSystems production staff to make sure that all elements meet the technical requirements for publication. They will contact you if anything needs to be revised before copyediting and production can begin. Otherwise, you will be notified when your proofs are ready to be viewed.

Publication Fees:

We recognize that the video files can become quite large, and so to avoid quality loss ASM suggests sending the video file via <https://www.wetransfer.com/>. When you have a final version of the video and the still ready to share, please send it to mSystems staff at mssystemsjournal@msubmit.net.

For mSystems research articles, if you would like to submit an image for consideration as the Featured Image for an issue, please contact mSystems staff at mssystemsjournal@msubmit.net.

Sincerely,

Youjun Feng
Editor, mSystems

Journals Department
Figure S4: Accept
Table S2: Accept
Dataset S1: Accept
Figure S2: Accept
Figure S1: Accept
Figure S5: Accept
Figure S3: Accept
Table S1: Accept
Dataset S2: Accept
Figure S6: Accept

Comment

The authors have fully answered all the questions raised previously, and honestly admitted that some mechanisms are not fully explored, which need further study. Moreover, the authors also have changed the manuscript according to the questions raised by the reviewers. So my concerns are adequately addressed, and I agree to accept this article.